# Contrasting Effects of Local Environmental and Biogeographic Factors on the Composition and Structure of Bacterial Communities in Arid Monospecific Mangrove Soils

T. Thomson,[a,b] M. Fusi,[b,c] M. F. Bennett-Smith,[b] N. Prinz,[a] E. Aylagas,[b] S. Carvalho,[b] C. E. Lovelock,[d] B. H. Jones,[b] J. I. Ellis[a,b]

aUniversity of Waikato, School of Science, Tauranga, New Zealand
bKing Abdullah University of Science and Technology (KAUST), Biological and Environmental Sciences and Engineering Division (BESE), Thuwal, Saudi Arabia
cSchool of Applied Sciences, Edinburgh Napier University, Edinburgh, United Kingdom
dSchool of Biological Sciences, The University of Queensland, St Lucida, Australia

**ABSTRACT** Mangrove forests are important biotic sinks of atmospheric $CO_2$ and play an integral role in nutrient-cycling and decontamination of coastal waters, thereby mitigating climatic and anthropogenic stressors. These services are primarily regulated by the activity of the soil microbiome. To understand how environmental changes may affect this vital part of the ecosystem, it is key to understand the patterns that drive microbial community assembly in mangrove forest soils. High-throughput amplicon sequencing (16S rRNA) was applied on samples from arid *Avicennia marina* forests across different spatial scales from local to regional. Alongside conventional analyses of community ecology, microbial co-occurrence networks were assessed to investigate differences in composition and structure of the bacterial community. The bacterial community composition varied more strongly along an intertidal gradient within each mangrove forest, than between forests in different geographic regions (Australia/Saudi Arabia). In contrast, co-occurrence networks differed primarily between geographic regions, illustrating that the structure of the bacterial community is not necessarily linked to its composition. The local diversity in mangrove forest soils may have important implications for the quantification of biogeochemical processes and is important to consider when planning restoration activities.

**IMPORTANCE** Mangrove ecosystems are increasingly being recognized for their potential to sequester atmospheric carbon, thereby mitigating the effects of anthropogenically driven greenhouse gas emissions. The bacterial community in the soils plays an important role in the breakdown and recycling of carbon and other nutrients. To assess and predict changes in carbon storage, it is important to understand how the bacterial community is shaped by its environment. Here, we compared the bacterial communities of mangrove forests on different spatial scales, from local within-forest to biogeographic comparisons. The bacterial community composition differed more between distinct intertidal zones of the same forest than between forests in distant geographic regions. The calculated network structure of theoretically interacting bacteria, however, differed most between the geographic regions. Our findings highlight the importance of local environmental factors in shaping the microbial soil community in mangroves and highlight a disconnect between community composition and structure in microbial soil assemblages.

**KEYWORDS** microbiome, 16S rRNA, microbial biogeography, ecological processes, community structure, co-occurrence network analysis, community assembly

Address correspondence to T. Thomson, timi.thomson@gmail.com.

The authors declare no conflict of interest.

Mangrove forests are among the world's most efficient natural "Blue Carbon" sinks (1). About three-quarters (76.5%) of the carbon bound in mangrove forests is stored in the soil (1, 2). This globally important ecosystem service is mainly driven by

the activity of the associated soil microbial communities (3–6). In particular, mangrove soil bacteria comprise up to 30% of the soil biomass and play a pivotal role in regulating the forest's biochemical processes (7, 8). Yet, despite their role in carbon sequestration and additional ecosystem services microbial communities have not received the same attention as those in other environments such as coral reefs, the pelagic ocean, forests, and agricultural pastures (9–12).

Mangrove soils are characterized by high spatial heterogeneity of their physical, chemical, and biological components (13). Notably, leaf and root litter are hot spots of microbial activity due to decomposition (14–16). Oxygen is usually confined to the uppermost millimeters of the soil profile, but the presence of aerial roots and burrowing organisms can introduce oxygen into deeper layers (17–19). Such settings create microhabitats that support a wide range of microbial assemblages (20). On a wider spatial scale, mangrove forests display a distinct zonation between the seaward edge where tall fringing forests occur, and the interior of the forest where shrub forms of mangrove often dominate (21). Tree height, nutrient availability, productivity, bioturbation, salinity, and hydrology create contrasting settings between these zones, greatly influencing the physicochemical parameters of the soil (13, 17, 22–25). Studies on small-scale distributions (centimetres to meters) of microbial communities along environmental gradients (e.g., bioturbated soil by animal burrowing or plant root growth) have identified high levels of community variability, thereby confirming the complex patchwork of microbial assemblages in mangrove soils (19, 26–28). Despite this high intraforest variation, comparisons of microbial communities of mangrove soils between forests (on regional and global scales) yielded high levels of similarity (29–31). This emphasizes the importance of the local environmental conditions in determining the soil microbiome which have received increased attention recently (32–35). However, there is still limited research on the variation of these factors on a larger scale and little is known about the factors that influence bacterial community assembly in mangrove soils (36).

In community ecology, the nature and mechanisms of species distributions are widely studied and have largely been adapted by the field of microbial ecology (37, 38). It is now widely agreed that microorganisms follow biogeographic patterns similar to those of macroorganisms, and are limited by a combination of historical and geographic/environmental settings (37–41). This framework of community assembly recognizes four key processes that shape ecological communities that can be of stochastic or deterministic nature: selection, drift, speciation, and dispersal (37). Extrinsic processes on various spatial scales that are associated with species pools and dispersion influence community composition as do intrinsic environmental and biotic processes (42). As soil microbial communities of mangrove forests facilitate ecosystem processes of global importance, understanding the mechanisms that shape microbial community composition can aid in predicting changes resulting from fluctuating environmental and climatic conditions (12, 39). While there have been recent efforts to better understand microbial community assembly in both coastal and mangrove sediments (36, 43), a novel aspect of this study is the use of a hierarchical study design that takes into account various spatial scales.

Recent work indicates that, in order to ensure the functioning of an ecosystem, the structure of its microbial community is as important as its composition (44–47). Throughout this paper, we refer to community structure as network topology characteristics that imply interactions within the community, and we define community composition as alpha and beta diversity.

Network analyses of co-occurring microbial taxa have proven useful in the characterization of microbial community structures, despite the limitations of this statistical method, providing a solid sampling design and sufficient replication is considered (46, 48, 49). Disentangling patterns of connectedness between members of the microbial community and their spatial variability is an important step toward understanding the importance of single members or groups of the microbial community in facilitating

functional diversity and resilience (50). The functional diversity of a system describes the traits of organisms present within it that potentially influence or contribute to changes in this system (51). Ecological and functional microbial redundancies are key drivers of ecosystem resilience (49). As mangroves are threatened by numerous factors (e.g., sea level rise, overexploitation, drought, increased/decreased salinity) (52–54), patterns of microbial biodiversity can provide insights on the effective capacity of the mangrove to buffer these changes.

Based on the evidence above, we hypothesized that local environmental parameters are important controlling factors of the soil bacterial community, which may exceed the importance of geographic region. Therefore, we expected soil bacterial communities associated with the same species of mangroves and under similar climate conditions to vary more on local scales compared to variation at global scales. Moreover, physico-chemical conditions in different forests and between forest zones will have a significant influence on soil bacterial networks and the keystone taxa within them.

To test our hypotheses, we compared the bacterial community of two monospecific *Avicennia marina* mangrove forests over various spatial scales using 16S rRNA gene sequencing, aiming to identify the selective forces that determine the bacterial assembly patterns within the microbiome in this highly specialized environment. We analyzed the bacterial communities at different depths in the soil, distance from the sea, in forests with different exposures to oceanic influences, and from two distant geographic regions that are both dominated by a hot and arid climate (i.e., Saudi Arabia and Australia) (55).

## RESULTS

**Diversity of bacterial communities.** The rank-abundance relationship displayed in Fig. 1A shows the disproportional abundance of rare amplicon sequence variants (ASVs) (high rank values) compared to few highly abundant ASVs in the community, which is consistent across all sampling sites. Notably, the subsurface samples from the shrub zone show the steepest decline along the ranks, indicating fewer highly prevalent ASVs in these samples. Other than that, no definite trends are obvious between the samples.

The species richness differed significantly for each combination of geographic region, exposure, and zone but not for depth (geographic region × exposure × zone; $F_{1,103}$ = 8.83, *P value* = 0.004; Fig. 1B, Table S3). Species richness (observed ASVs) generally showed a significantly higher number of ASVs in Saudi Arabia compared to Australia (TukeyHSD, $P = 0.002$; 16.4% increase). In Australia, species richness was similar between the zones of the exposed site, while it varied significantly between zones of the sheltered site (TukeyHSD, $P = <0.001$; 121.7% higher in the fringe) where the richness was higher in the soils of the tall fringing forest. In Saudi Arabia, species richness was generally higher in the shrub than in the fringe (TukeyHSD, $P = 0.02$; 17.4% increase).

The variation of the Shannon diversity index was only significant between the interaction of local factors (zone × depth; $F_{1,104}$ = 5.28, *P value* 0.02; Table S4). Except for the exposed site in Australia, the differences between depths in the soil were significant in the shrub zone of the forest (TukeyHSD, $P < 0.001$; 8.9% higher in the surface), with higher diversity scores in the surface layers of the soil (Fig. 1C).

The differences in phylogenetic diversity were significant between the factors geographic region, exposure, and zone (geographic region × exposure × zone; $F_{1,103}$ = 14.66, *P value* < 0.001; Table S5). Phylogenetic diversity was highest in the shrub of Saudi Arabia and generally showed higher values for the shrub zone compared to the fringe (TukeyHSD, $P < 0.001$; 25.1% increase) (Fig. 1D). As with the Species richness and the Shannon diversity index, the phylogenetic diversity of the sheltered site in Australia showed an opposing trend between the zones, yielding higher diversity in the fringe compared to the shrub.

**Composition of bacterial communities.** The proportion (throughout the document referred to as relative abundances) of phyla between factors are shown in Fig. 2A and Table S6. *Proteobacteria* was the most abundant phylum across all samples

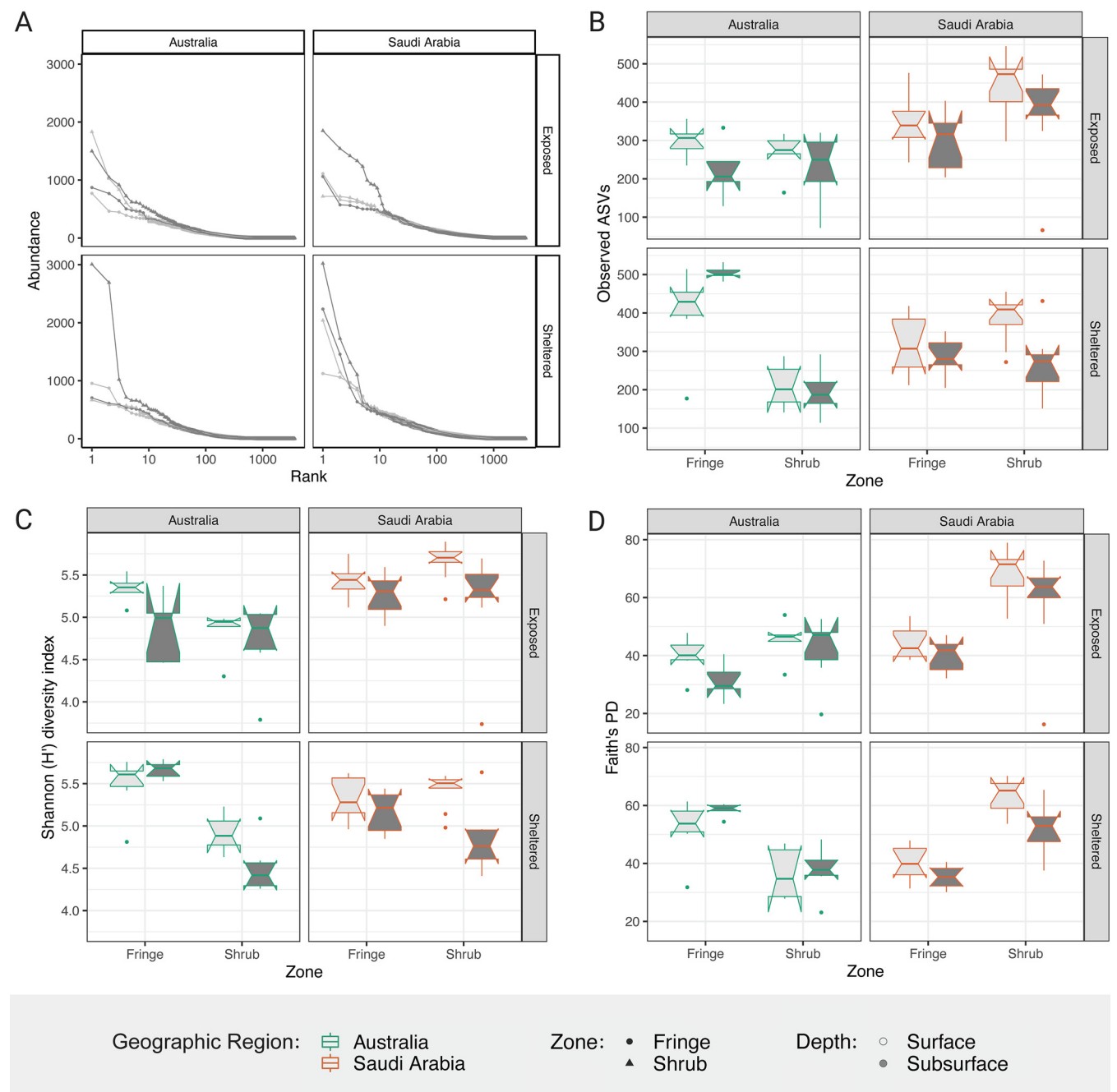

**FIG 1** Within sample diversity of bacterial communities from arid mangrove soils. Alpha diversity measures of the bacterial communities detected in mangrove soils from Australia (green outline) and Saudi Arabia (orange outline) across zones (fringe, shrub) and depths (surface in light gray, subsurface in dark gray). (A) Rank-abundance relationship of ASVs by location. (B) Species richness described by the number of observed ASVs. (C) Shannon diversity index. (D) Phylogenetic diversity index (Faith's PD). The boxplots indicate the median with the interquartile range (IQR) between the 25th and the 75th percentile and the whiskers extend 1.5*IQR. Boxes were plotted with the notch (+/- 1.58*IQR/sqrt[n]) to display likely statistical significance if notches do not overlap.

(58.96%), followed by *Bacteroidetes* (17.44%), *Chloroflexi* (8.06%), *Calditrichaeota* (3.81%), and *Nitrospirae* (1.59%), contributing to 90% of the total community. Within *Proteobacteria*, the most abundant class was *Deltaproteobacteria* (31.60%), followed by *Gammaproteobacteria* (23.34%), *Bacteroidia* (13.65%), *Anaerolineae* (7.12%), and *Alpha-proteobacteria* (4.44%) (Table S7). In the shrub zone, the differences in relative abundance of phyla between surface and subsurface was more obvious than in the tall fringing zone. This is consistent for exposed and sheltered sites in both geographic regions. Especially in the shrub samples, the relative abundances of *Proteobacteria* was

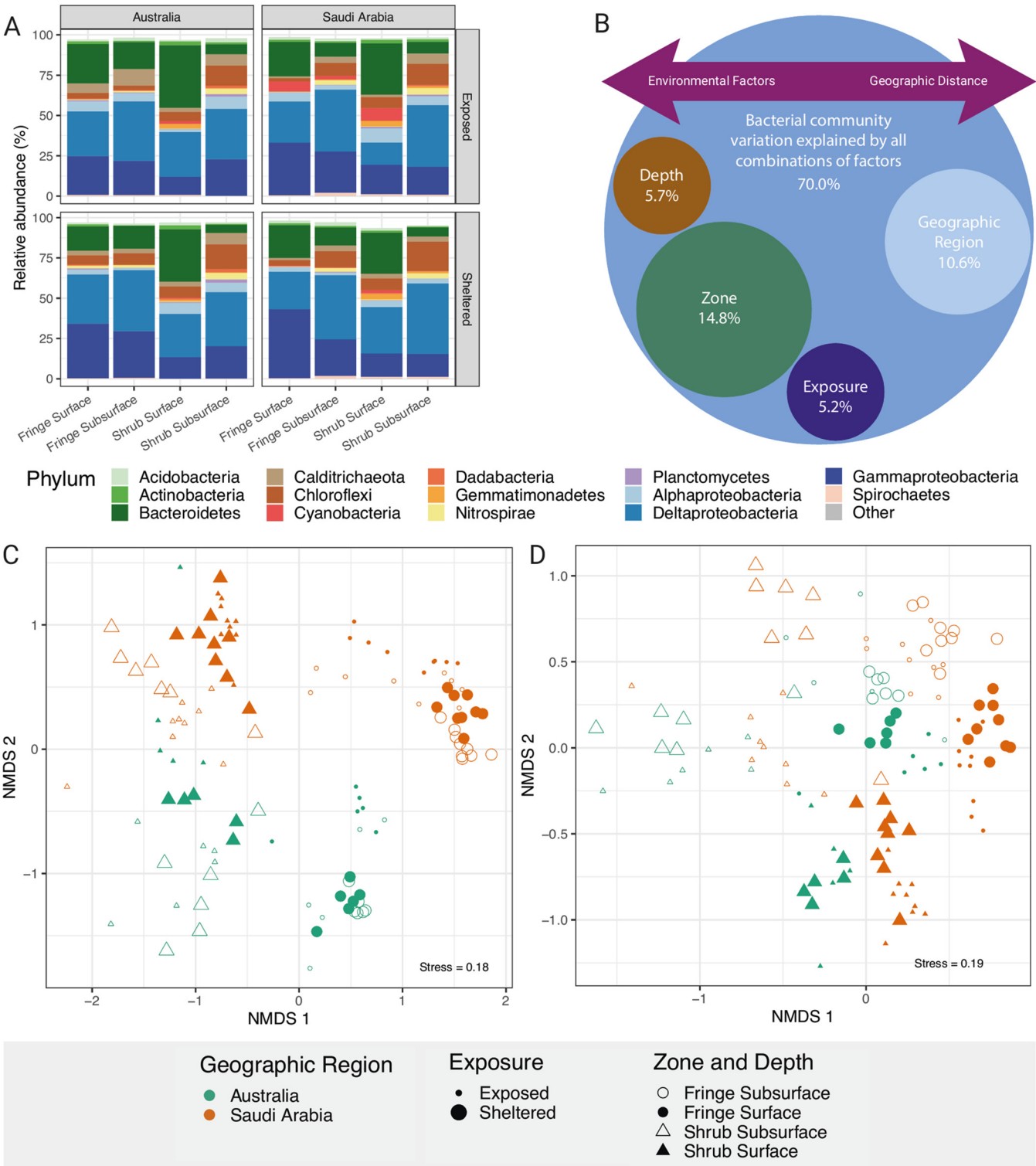

**FIG 2** Comparison of different bacterial communities between the experimental factors geographic region, exposure, zone, and depth. (A) Relative abundance of phyla across the whole data set. The top 12 most prevalent phyla were displayed and the remaining grouped as "Other." The most abundant classes within the *Proteobacteria* were additionally included (blue colors). (B) Schematic representation of the R2-values from the PERMANOVA. The large bubble represents the proportion of variation explained by the interaction of all experimental factors. The small bubbles show the proportion of variation explained by each individual factor. The arrow above indicates the environmental factors influencing the separation. (C) Nonmetric multidimensional scaling (NMDS) plots of the community matrix across experimental factors at ASV level, and (D) at genus level across geographic region, exposure, zone and depth.

higher in the subsurface layer than at the surface (Fig. S3). *Deltaproteobacteria* were more abundant in the subsurface of all samples, while *Gammaproteobacteria* were generally more abundant in the fringing zone (Fig. 2A). The second most abundant phylum, *Bacteroidetes*, was more abundant in the surface layer of all locations (Fg. S4). *Chloroflexi*, *Calditrichaeota*, and *Nitrospirae* were more abundant in the subsurface layer, except from the fringe samples of the sheltered site in Australia (Fig. S5 to S7). *Cyanobacteria* were abundant at the surface and more so in Saudi Arabian samples. The relative abundances of *Cyanobacteria* were particularly high in both zones of the exposed site in Saudi Arabia with the exposed shrub of Australian samples also showing slightly elevated abundances (Fig. S8).

The NMDS plot (Stress = 0.18), calculated from the community matrix at ASV level, separates the samples by the factor zone along the first axis (Fig. 2C). The factor geographic region mainly accounts for the spread of the data points along the second axis. This is more obvious for the fringe samples, whereas the samples from the shrub overlap slightly, leading to a slight overlap of communities from different geographic regions. The factors exposure and depth fail to separate the samples into distinguishable groups. At genus level (Stress = 0.19), the factor zone still separated the samples; however, the factor geographic region did not (Fig. 2D). Instead, the factor depth was able to help separate the data in the ordination space, while the factor exposure did not. The models calculated from the CAP analysis were able to match the bacterial communities with the correct factor combination between 80 – 100% of the time at ASV level and genus level (13 and 11 perfect scores out of 16, respectively). The model at phylum level performed poorly (between 44.4 – 100%), which is why no further analysis was carried out at a higher taxonomic level.

PERMANOVA showed a significant interaction on the taxonomic community composition at ASV level across all factors (geographic region $\times$ exposure $\times$ zone $\times$ depth; $F_{1,114} = 3.36$; *P value* = 0.001; Table S8). A pairwise comparison of the significant four-way interaction term confirmed the statistical significance between ecologically relevant combinations of the factors geographic region, exposure, zone, and depth (Table S9). A significant interaction term between geographic region, exposure, zone, and depth was found at genus level (geographic region $\times$ exposure $\times$ zone $\times$ depth; $F_{1,114} = 3.99$; *P value* 0.002; Table S10). The variation within the data set explained by the individual factors was 10.6% by geographic region, 5.2% by exposure, 14.8% by zone, and 5.7% by depth (Table S8). All combinations of factors were able to constrain 70.0% of variation within the data set (Table S8).

**Network analysis.** The network analysis revealed distinct co-occurrence patterns between bacterial ASVs of mangrove soils across the factors of this study. The Australian networks contained a lower average number of nodes and a higher number of edges than Saudi Arabia, resulting in a higher density and average number of neighbors, but lower network diameter. The modularity, as indicated by the number of modules detected in each network, was higher in Saudi Arabia, whereas the clustering coefficient and the centralization was higher in Australian networks (Table 1, File S2).

The betweenness centrality metric per phylum showed differences between the experimental factors (Fig. 3A). The betweenness scores in Saudi Arabia seemed generally to be more evenly distributed between phyla than in Australia. This was driven by the high variability of *Calditricaeota*, *Planctomycetes*, and *Spirochaetes* in Australian samples. Between zones, *Cyanobacteria* and *Zixibacteria* showed distinct differences. *Cyanobacteria* were more central in shrub samples of both geographic regions and *Zixibacteria* were also more central in the shrub samples. Between depths, *Cyanobacteria* and *Nitrospirae* varied. Values for *Cyanobacteria* were higher in the surface layers and *Nitrospirae* were more central in the subsurface of most samples. *Actinobacteria*, *Gemmatimodetes*, and *Chloroflexi* showed consistently high betweenness centrality values, while values for *Proteobacteria* and *Bacteroidetes* were constant but average in all samples.

The topological coefficient highlighted the role of yet different phyla within the networks (Fig. 3B). As for the betweenness centrality, the values for the topological coefficient

**TABLE 1** Summary statistics characterizing the bacterial co-occurrence networks of all sites in Australia and in Saudi Arabia

| Geographic region | Exposure | Zone | Depth | Nodes | Edges | Edges per node | Clustering coefficient | Centralization | Diameter | Avg neighbors | Density | No. of modules |
|---|---|---|---|---|---|---|---|---|---|---|---|---|
| Australia | Exposed | Fringe | Surface | 609 | 14793 | 24.29 | 0.58 | 0.52 | 7 | 48.58 | 0.08 | 6 |
| | | | Subsurface | 609 | 14793 | 24.29 | 0.58 | 0.52 | 7 | 48.58 | 0.08 | 6 |
| | | Shrub | Surface | 451 | 6504 | 14.42 | 0.70 | 0.11 | 9 | 28.84 | 0.06 | 14 |
| | | | Subsurface | 597 | 8333 | 13.96 | 0.70 | 0.60 | 7 | 27.92 | 0.05 | 5 |
| | Sheltered | Fringe | Surface | 495 | 5808 | 11.73 | 0.62 | 0.67 | 6 | 23.47 | 0.05 | 5 |
| | | | Subsurface | 733 | 6323 | 8.63 | 0.44 | 0.40 | 7 | 17.25 | 0.02 | 6 |
| | | Shrub | Surface | 675 | 2024 | 3.00 | 0.45 | 0.02 | 21 | 6.00 | 0.01 | 17 |
| | | | Subsurface | 418 | 5530 | 13.23 | 0.55 | 0.36 | 6 | 26.46 | 0.06 | 3 |
| Saudi Arabia | Exposed | Fringe | Surface | 423 | 9642 | 22.79 | 0.52 | 0.45 | 7 | 45.59 | 0.11 | 3 |
| | | | Subsurface | 736 | 5387 | 7.32 | 0.33 | 0.16 | 10 | 14.64 | 0.02 | 10 |
| | | Shrub | Surface | 737 | 6088 | 8.26 | 0.37 | 0.26 | 7 | 16.52 | 0.02 | 10 |
| | | | Subsurface | 769 | 5514 | 7.17 | 0.37 | 0.29 | 8 | 14.34 | 0.02 | 8 |
| | Sheltered | Fringe | Surface | 726 | 5569 | 7.67 | 0.32 | 0.43 | 10 | 15.34 | 0.02 | 7 |
| | | | Subsurface | 661 | 4178 | 6.32 | 0.42 | 0.24 | 11 | 12.64 | 0.02 | 9 |
| | | Shrub | Surface | 609 | 4812 | 7.90 | 0.38 | 0.22 | 8 | 15.80 | 0.03 | 7 |
| | | | Subsurface | 736 | 5725 | 7.78 | 0.30 | 0.31 | 11 | 15.56 | 0.02 | 9 |

were more evenly distributed between phyla in Saudi Arabian samples. Highly variable in Australian samples were *Actinobacteria* and *Cyanobacteria*. *Gemmatimodetes* showed high values in the soils of sheltered sites of Australian forests. Across zones, *Spirochaetes* and *Zixibacteria* showed the strongest differences with generally higher scores in the fringe. Between depths, *Spirochaetes* showed the greatest differences with generally slightly higher values in the Surface. Overall, *Nitrospirae*, *Spirochaetes*, *Zixibacteria*, and in some samples *Gemmatimodetes* had the highest scores. Keystone taxa analysis showed an increased number of important nodes in Saudi Arabian compared to Australian networks (Fig. S9).

In Australian networks, the relative proportion of nodes at higher degrees

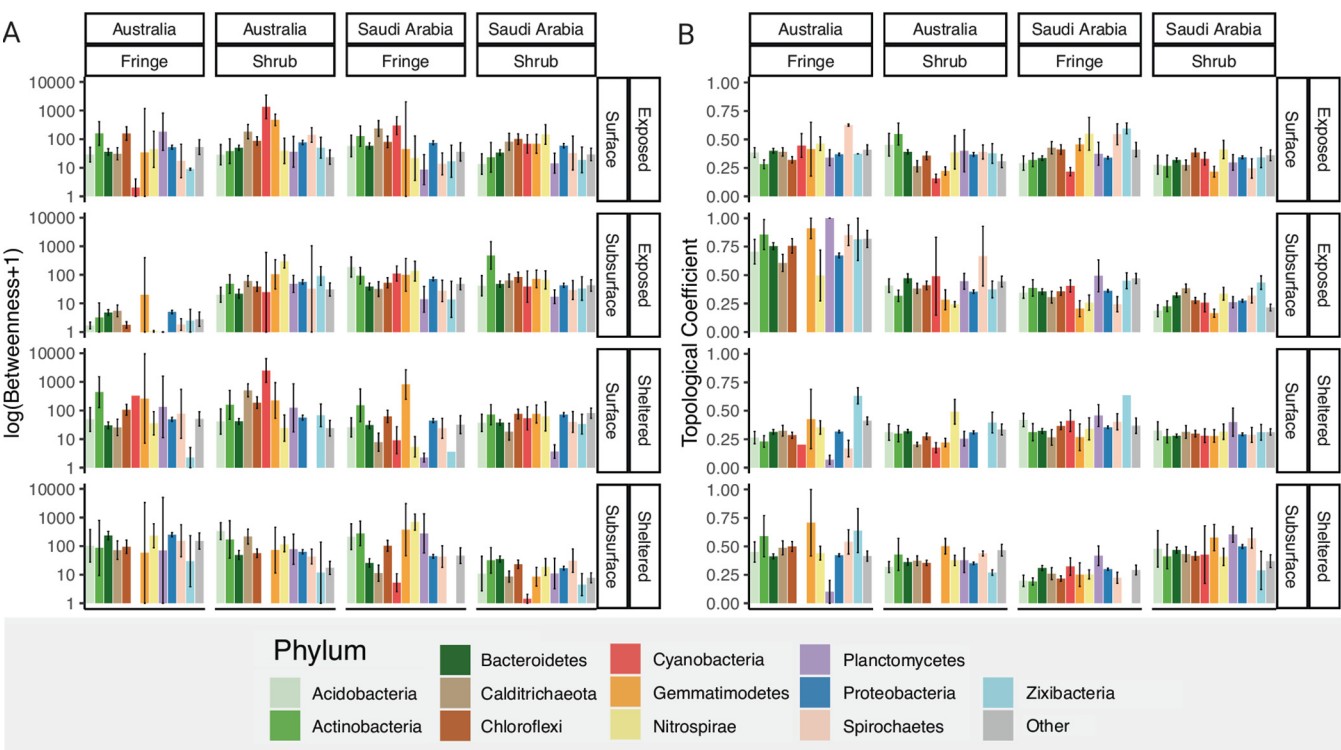

**FIG 3** Selected network metrics by phylum. (A) Betweenness centrality and (B) Topological coefficient. The 12 most prevalent phyla (within the networks) were selected and the remaining grouped as "Other". Mean scores are displayed with standard errors calculated by the stat_summary function in ggplot2. Bars without standard errors display single occurrences.

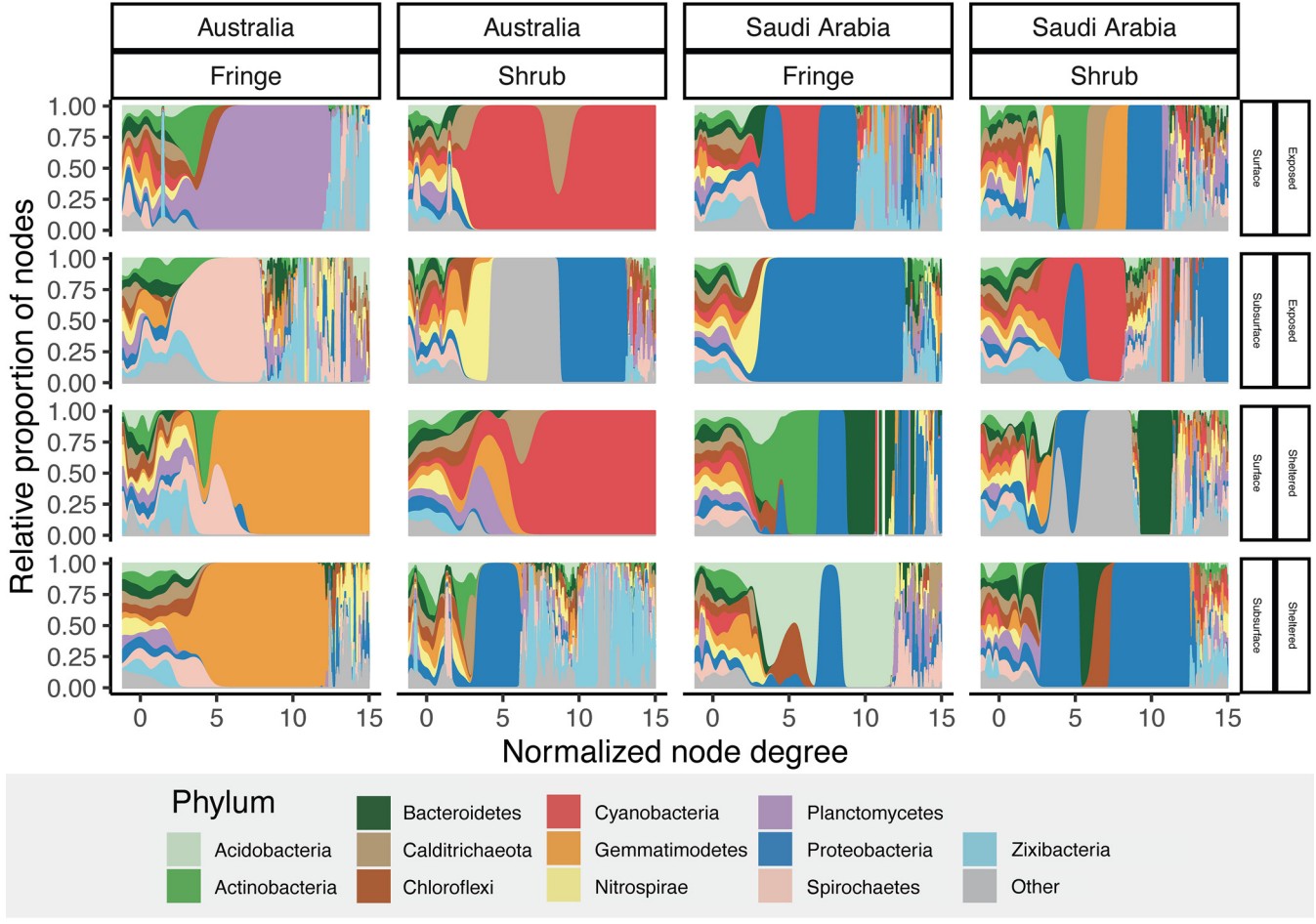

**FIG 4** Comparison of high degree scores in co-occurrence networks. Density plots of the relative proportion of nodes against the normalized node degree colored by phylum across geographic region, exposure, zone and depth. The proportion of nodes is calculated by the Kernel-Density function.

(normalized) was dominated by single phyla, especially in the shrub forests (Fig. 4). In the surface samples of the shrub in Australia, *Cyanobacteria*, and at a lower node-degree *Calditrichaeota*, were the only phylum with normalized node-degrees higher than 5, whereas in the subsurface of the same samples *Proteobacteria*, *Zixibacteria*, and less abundant phyla (grouped as *"Other"*) filled this role. In the surface samples of the fringe in Australia, *Planctomycetes*, *Chloroflexi*, and *Actinobacteria* were most connected in exposed forests, and *Gemmatimodetes* in the sheltered forests. This stands in a strong contrast to the Saudi Arabian sites, in which nodes with high degrees were shared between different phyla, thereby representing a higher diversity of highly connected nodes.

**Functional assignments.** Across all factors, "respiration of sulfur compounds" was the most prominent function, with an increased presence in the subsurface layer (Fig. 5). "Chemoheterotrophy" was an abundant pathway with higher prevalence in the surface layer, especially at the sheltered site in Saudi Arabia. Functional traits related to photosynthesis, such as "cyanobacteria" (grouped as a function by FAPROTAX), "phototrophy," "photoautotrophy," and "oxygenic photoautotrophy" were more abundant at the surface layers, most notably in the exposed site in Saudi Arabia. Generally, these functions were more abundant in the exposed site compared to the sheltered site. "Fermentation" was a prominent function in the sheltered site in Saudi Arabia, as well as the fringe samples of the other sites. "Aerobic nitrite oxidation" and "nitrification" were more abundant in the subsurface compared to the surface. The variation between all factors were statistically significant (geographic region × exposure × zone × depth; PERMANOVA;

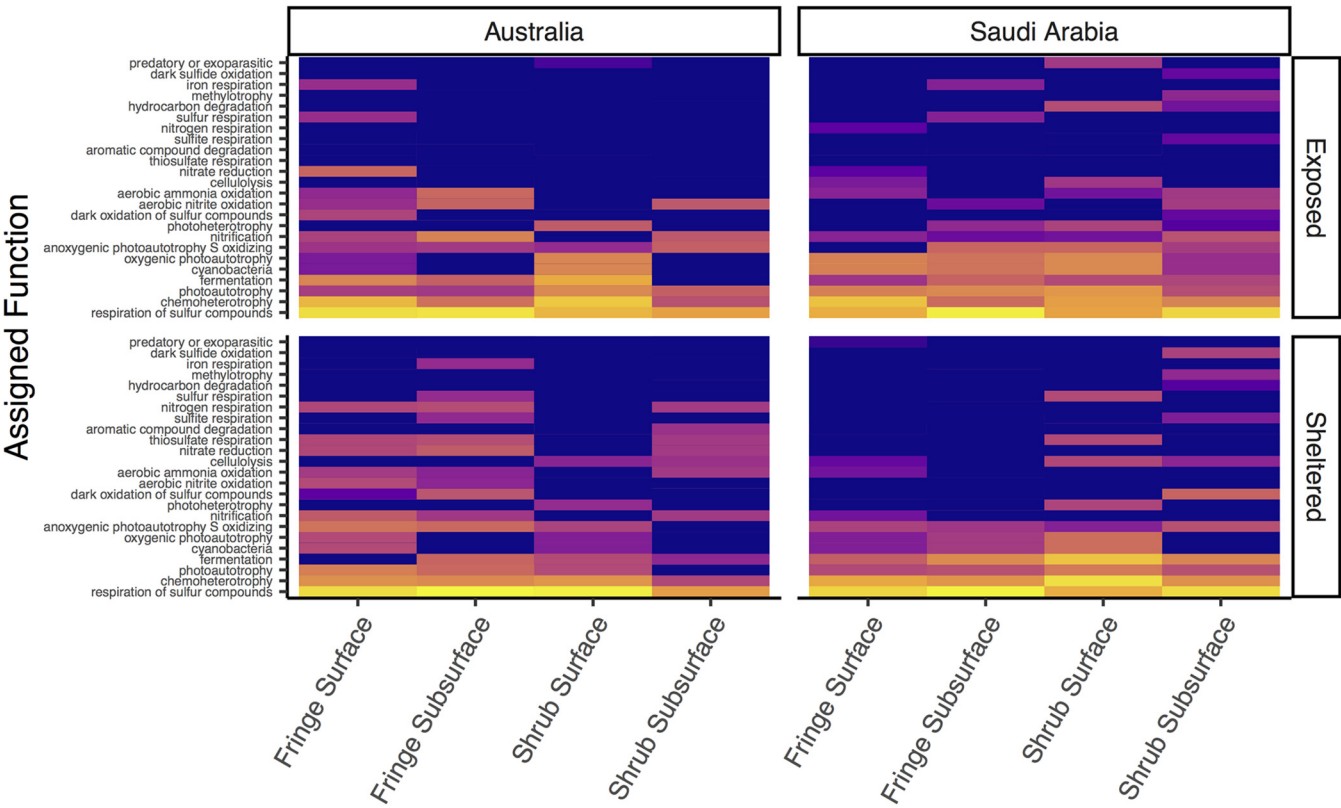

**FIG 5** Theoretical functional assignments. Heat map showing the distributions of bacterial functions that were assigned by FAPROTAX across geographic region, exposure, zone and depth. Values were log-transformed with lighter values indicating higher abundances.

permutations = 999, method = bray, $F_{1,114}$ = 5.79; *P value* = 0.001; Table S11), underlining the variability of the inferred function across factors.

## DISCUSSION

Environmental factors characterizing distinct areas within arid *Avicennia marina* mangrove forests were more important in shaping the bacterial soil community than the geographic region of the forest based on community composition analysis. We found that the bacterial communities of mangrove soils differed significantly between geographic regions and local factors based on alpha- and beta-diversity. Furthermore, they were compositionally less similar between contrasting zones within a forest, than between different forests on a continental scale. In contrast, the hypothetical interaction structure of mangrove soil bacteria, as represented by co-occurrence networks, was strongly associated with the geographic region. This finding highlights potentially important differences between the composition and the co-occurrence structure of the microbial community. Consequently, the role of local environmental, and global factors is discussed in turn.

**Local factors: the role of environmental variability.** Mangrove forests are spatially heterogeneous along distinct intertidal and depth gradients within which the physicochemical conditions can differ dramatically (25, 56). In several studies on terrestrial and coastal soil microbiomes across different latitudes and ecosystems, environmental parameters (especially pH) were more influential on the community composition than geographic distance (11, 36, 43, 57). Similarly, a continuity of the microbial community driven by environmental factors has also been proposed for the pelagic

ocean, as well as for brackish water bodies (58, 59). Our results confirm a separation of the bacterial community following the structure of the forest, as has been suggested previously (60). The increased potential of dispersal at the fringe by freely moving water masses and the increased cycle of surface particle sedimentation and resuspension may lead to a homogenization of the bacterial communities in this zone (61, 62). The importance of tidal mixing for homogeneity of microbial communities is consistent with an earlier study that found the microbial soil community at the fringe of a mangrove soil to be more similar to that of the adjacent mudflat, than to the interior of the same forest (63). The number of observed ASVs increased significantly in the shrub of Saudi Arabian forests, as did the phylogenetic diversity, thus supporting the hypothesis of higher diversity away from the homogenizing forces of the fringe environment (64–66). In Australia these differences were less pronounced in the exposed site but opposite in the sheltered site with higher numbers of observed ASVs in the fringe compared to the shrub. This discrepancy may be due to differences in stressors affecting forests in both geographic regions such as strong tidal flow in Australia and high salinity or nutrient scarcity in Saudi Arabia (67).

Most notable was the difference in variation of alpha diversity metrics between depths of the fringe and the shrub. While the respective diversity metric was fairly consistent between depths in the fringe, it varied more strongly in the shrub, generally showing higher values at the surface (68). The higher variability between depths in the shrub was similarly highlighted by the relative abundance of phyla between sites indicating a more diverse bacterial community in shrub soils compared to the soils of tall fringing mangroves. The ordination analysis also shows this diversification in the shrub samples of both geographic regions. While the samples from the fringe grouped together tightly, the shrub samples show more heterogeneity in their spatial arrangement. All these diversity measures clearly show an increased diversification of the bacterial community in the shrub forest, regardless of its geographic location, which can be explained by the reduced hydrodynamic forcing in the high intertidal (69). Selective forces in habitats in which dispersal limitation prevails may be the driving factors of this diversification as has been discussed in other studies (39, 40, 70). The differing results at ASV compared to genus level indicate a potential impact of local environmental and evolutionary factors on bacterial community assembly over geographic distance. At ASV level the geographic region was the second strongest explanatory factor (Fig. 2C; Table S8), while it lost its explanatory strength when analyzed at genus level. Instead, the local factor depth became more important in separating the data in ordination space (Fig. 2D; Table S10). This indicates a similarity at a higher taxonomic level (genus), which diversifies differently between the geographic regions at the finest taxonomic scale (ASV). Such various levels of taxonomic resolution have been shown to change the level of differentiation between bacterial communities (63, 71), which has been hypothesized to be due to evolutionary factors such as genetic drift and/or speciation (40). The importance in acknowledging the taxonomic resolution (or taxonomic depth) when searching for biogeographic patterns has been highlighted before (72, 73) and may be an important factor in disentangling mechanisms driving microbial community assembly (40).

**Global patterns: distance and historical contingency.** While the local environmental factors were dominant in shaping the bacterial community composition, our results also provide evidence for differences between the bacterial communities of geographically distant mangrove forests. These findings contradict the traditional paradigm of "everything is everywhere" (74), which supports the theory of microbiological cosmopolitanism (75), in favor of the contemporary approach of an interplay of contrasting assembly mechanisms (37–40). The higher number of observed ASVs in Saudi Arabia, which is supported by higher phylogenetic diversity, may stem from the less favorable conditions for growth such as oligotrophy, higher levels of carbonates, high salinity, and extreme temperatures in the Red Sea (76, 77). These conditions along with a small tidal range (0.2 m) in Saudi Arabia which may limit dispersal (78, 79), can

enhance the effects of environmental selection. Because we did not observe strong separation of bacterial communities at genus level among geographic regions suggests that evolutionary processes such as genetic drift or speciation may be in place (40). Speciation is considered to have a negligible effect in communities that are connected via dispersal (61, 80, 81), but may influence bacterial communities that are separated by environmental boundaries (82–84).

**Network analysis.** The results from our co-occurrence network analysis show that community composition is not directly linked to network structure, as the networks clearly differ more among the geographic regions compared to local factors. In particular, the bacterial networks in Australian soils appear smaller, yet more closely connected in comparison with Saudi Arabian networks. The lower modularity and higher centralization values of the Australian networks suggest a network structure that is built around a few well-connected nodes. In contrast, in Saudi Arabia, the networks support a larger number of nodes that hold a central position (decentralized), with more independent functional modules. The difference in network structure among geographic regions may be due to the generally nutrient poor conditions, high water temperatures, and seasonal fluctuations (leading to long periods of desiccation) in Saudi Arabia (77) which may create an environment that promotes the formation of multiple functionally isolated groups of bacteria.

Important phyla for the connectedness of the networks were identified, including *Cyanobacteria*, *Gemmatimodetes*, *Zixibacteria*, and *Spirochaetes*. The high betweenness values of *Cyanobacteria* in surface samples suggests the possible importance of this phylum in regulating the occurrence or absence of clusters of co-occurring bacteria within soils. Such a role has previously been suggested for *Cyanobacteria* in different environments, for example in desert soils (85). The regulating role of *Cyanobacteria* may be due to their ability to synthesize C- and N-rich organic compounds in nutrient limited conditions (86), or the formation of tightly connected mats which can select for or inhibit other bacteria (87). Their importance in the surface samples of the shrub in Australia was particularly pronounced, where they were the only phylum with a high node degree score (Fig. 4). This stands in contrast to the relative abundance of *Cyanobacteria* which is higher in Saudi Arabia than in Australia (Fig. S8). In Saudi Arabia resource scarcity may have supported high relative numbers of *Cyanobacteria* in the surface layers of the sediment but in Australia they were more important in structuring the network topology.

The dominant role of the phylum *Gemmatimodetes* in facilitating connections between and within modules (as shown by high betweenness and high topological coefficient) is difficult to interpret since limited physiological information on this Phylum is available (88). *Gemmatimodetes* were first described in 2003 (89), even though they are routinely detected in culture independent sequencing approaches and comprise about 2% of terrestrial soil communities (90). However, they have been proposed as oligotrophs with the potential of degrading complex organic material (OM), which is often present in high concentrations in mangrove soils (91). Their role as seen in the network structure may therefore be the degradation of more complex carbon substrates into smaller fractions, making these readily available for modules of bacteria to utilize. *Zixibacteria*, also a recently described phylum (92), were well connected within modules of many networks, as indicated by high topological coefficient scores. The metabolism of *Zixibacteria* has been suggested as highly versatile and adaptable to fluctuating environmental conditions (92, 93). The identification of genes potentially encoding nitrite/nitrate oxidoreductase (NXR), an important enzyme of the nitrification pathway, that has also been proposed to function as a nitrate reductase in anammox organisms, highlights a potentially important role of *Zixibacteria* in the nitrogen cycle of mangrove soils under aerobic or anaerobic conditions (92, 94). In deep sea sites of hydrocarbon seepage, *Zixibacteria* have been found to be the dominant fermenters of organic material, making OM more accessible for further degradation (95). The plasticity of metabolic pathways proposed for *Zixibacteria* may be the reason for

high co-occurrence rates of this phylum within tightly knit groups of other phyla within the network structure. A similar role of fermentation of OM may be played by bacteria of the phylum *Spirochaetes*, which have been shown to recycle necromass in polluted habitats, thereby providing substrates for further OM breakdown such as sulfate reduction (96, 97). Interestingly, the most abundant phylum across all samples, *Proteobacteria*, did not play an important role in the structure of the constructed co-occurrence networks. This was also the case for the next most abundant phylum *Bacteroidetes*. The phyla identified as creating the most important inter- and intramodule connectivity, were those with comparably low relative abundances such as *Zixibacteria*, *Gemmatimodetes*, and *Spirochaetes*. This finding underlines the rare taxa concept, that high abundance of a taxa is not a sign of its importance within the community (98, 99). Instead, taxa of low abundance but high connectedness can have a large influence on the facilitation of processes within the community (44, 100, 101). Overall, the topological parameters of the networks, separated by phylum, were unable to identify differences between the factors of this study.

The relative distribution of node degrees between phyla, which indicates the connectedness of phyla within the network, displayed distinct differences between the two geographic regions. In Australia, nodes with high degrees (>250) were almost exclusively dominated by a small number of phyla, indicating the disproportional importance of single phyla in those networks. In contrast, the Saudi Arabian soils featured a highly diverse assemblage of phyla contributing to the connectedness of the networks. This may be due to the strongly limiting conditions in Saudi Arabian mangroves, which require a diverse set of metabolic pathways to utilize the scarce resources present. The contrasting effects that different levels of stress can exert on microbial interactions have been discussed in the light of the intermediate stress hypothesis (102, 103) and the stress gradient hypothesis (67). In this study, it is likely that the nature of the local stressors differs between the two geographic regions, resulting in different profiles of community interaction.

**Functional implications.** The identified functional groups (assigned by FAPROTAX) differed mainly between local factors. "Respiration of sulfur compounds" was highly abundant in all sites. Metabolic pathways of the sulfur cycle are vital components of many marine sediments, including mangrove soils; sulfate respiration is one of the major respiration pathways in anoxic soils (104, 105), and is said to reduce methane emissions from intertidal soils by outcompeting methanotrophs in high salinity and sulfate-rich conditions (106, 107). "Chemoheterotrophy" was different between depths, probably separated along the soil profile by oxygen availability. The high abundance of chemoheterotrophs demonstrates the prominent heterotrophic bacterial community, which is supported by organic rich soils (108). Chemoheterotrophy has been described as a method of carbon recycling in microbial communities of marine soils (109). Chemoheterotrophic expression was notably higher in the surface of the shrub in Saudi Arabia compared to the Australian shrub sites, which may further suggest a more efficient reuse of the available resources in a system with low tidal exchange. Functions related to primary productivity, such as "photoautotrophy," "cyanobacteria" (grouped as a function by FAPROTAX), and "oxygenic photoautotrophy" were most different between the surface and the subsurface. Light is essential for phototrophy to take place, and most likely the main driver in this separation.

These functional assignments are based on the literature rather than gene expression or metabolic measurements and can only suggest possibly active functions within the sampled communities (110). In a comparative study such as this one, functional assignment can nevertheless be a relevant addition to the analysis of the bacterial communities and shed light on the contrasting metabolic pathways in different parts of the forest.

## CONCLUSION

Our findings suggest that the environmental parameters shaping the local soil environment are more important in determining the composition of the soil bacterial

community than geographic distance. However, this study also shows that geographic differentiation of the bacterial community is evident when investigated at the finest taxonomic resolution. The extent to which this variability is of functional importance needs to be assessed in future studies.

This work supports the importance of environmental selection as a driving process in microbial community assembly. Therefore, considering and accounting for local variability in environmental conditions, including the physical dynamics, is eminent when planning for mangrove restoration and scientific assessment of biogeochemical functions (e.g., carbon sequestration and nutrient processing). Understanding the spatial variability and distribution of microbial communities in mangrove soils, as well as their ecological functions, is important in order to predict changes and mitigate the impacts of intensified anthropogenic pressures and climate change (12, 20, 111, 112).

## MATERIALS AND METHODS

**Site descriptions and experimental design.** Sampling was conducted in two arid monospecific mangrove forests of *A. marina*, at two distant geographic regions: on the central west coast of Australia, near Exmouth (22.49° S; 114.33° E), and in the central Red Sea, on the western coast of Saudi Arabia, near Thuwal (22.33° N; 39.09°E) (Fig. 6). These regions lie approximately 9,500 km apart, on the southern and northern boundaries of the tropics, respectively, and share certain climatological characteristics, including low mean annual rainfall (249 mm and 54 mm, respectively) and high mean annual maximum temperatures (32.0°C and 36.3°C, respectively) (http://reg.bom.gov.au/climate/data/; http://www.saudi-arabia.climatemps.com/). The tidal range, however, differed substantially between the two geographic regions where the mean range in Australia (2 m) is about 1 order of magnitude higher than in Saudi Arabia (0.2 m). Sampling was conducted in June 2018 in Australia and in December 2018 in Saudi Arabia, hence during the winter season in both regions.

In both geographic regions, two contrasting sites were chosen to encompass variation in their hydrographic conditions. The sheltered sites were situated inside embayments that shielded them from the direct exposure to the open sea. The exposed sites face the open sea with direct exposure to wave/wind action and stronger oceanic influence. At each site, samples were collected from the tall fringe and dwarfed shrub zones of the mangrove forest, and each sample was split into two depth segments.

Six cores per zone were collected from Australian mangrove soils due to limited resources, while nine cores were sampled per zone in Saudi Arabia (as triplicates of each nested location; see Fig. 6). Cores were taken using a 60 ml polyethylene syringe with cutoff tip (3 cm diameter), that extended approximately 10 cm into the soil All cores were subsequently separated into surface (0 – 2 cm) and subsurface (5 – 7 cm) fractions. The soil depths at the exposed site in Saudi Arabia were very shallow (max. 5 cm), due to an underlying hard carbonate platform. In these environments, the deepest possible depth was sampled (3 – 5 cm). All samples were kept in the dark on ice and taken to the laboratory for storage at −80°C, immediately after the sampling was completed (2 - 3 h). Across geographic regions (Saudi Arabia/Australia), coastal exposures (exposed/sheltered), forest zones (fringe/shrub), and soil depths (surface/subsurface), a total of 120 samples were collected (of which 5 were later discarded during analysis).

**DNA extraction and target gene amplification.** DNA for bacterial community analysis was extracted from approximately 0.8 g of soil, using the DNeasy PowerSoil kit (Qiagen, Hilden, Germany). Using the specific primers 341F and 805R (113) with overhang illumina adapters (Illumina Inc., San Diego, CA, USA), the V3-V4 hypervariable regions of the 16S rRNA gene were amplified by PCR (PCR), at a final reaction volume of 25 $\mu$l per sample (114). The amplified samples were cleaned to remove primer-dimers and nontargeted DNA molecules using the SequalPrep Normalization Plate kit (Invitrogen, Carlsbad, CA, USA). Library preparation was carried out using the Nextera XT Index kit (Illumina Inc., San Diego, CA, USA) in combination with the Qiagen Multiplex PCR Master Mix (Qiagen, Hilden, Germany) that uses the HotStarTaq DNA polymerase. The cycler was set to 95°C as initial temperature for 15 min, followed by 8 cycles of 95°C for 30 s, 55°C for 90 s, and 72°C for 30 s, and a final extension with 95°C for 5 min (as specified by the manufacturer), before a second cleanup and normalization step was performed (SequalPrep Normalization Plate [96] kit, Invitrogen, Carlsbad, CA, USA). The libraries were then concentrated by vacuum centrifugation to approximately 12–15 nM, their concentration was measured by qPCR, and the amplicon length was validated with a Bioanalyzer at the KAUST Bioscience Core Lab. The samples were then sequenced on the Illumina MiSeq platform - v3 chemistry - (Illumina Inc., San Diego, CA, USA) with paired-end sequencing over 301 cycles.

**Bioinformatics.** The PCR primer sequences were removed from the dereplicated reads using the cutadapt software *v. 2.1* (115) with a max error rate of 7%. Untrimmed sequences were removed. The DADA2 software package *v. 1.10* (116) was used to differentiate exact amplicon sequence variants (ASVs) and remove chimeras. Filtering parameters used were: maxN = 0, maxEE = c (3, 6), trunclen = c (260, 190), trunQ = 2. DADA2's core algorithm models the errors in Illumina-sequenced reads and assigns ASVs with an accuracy of two base pairs (bp) difference, based on the quality score distribution (116). ASVs have been proposed as a new method to group Illumina sequencing reads according to exact matches rather than arbitrarily chosen similarity thresholds (116), providing reproducibility of marker gene-based studies, and simultaneously maximizing the biological variation that can be captured in

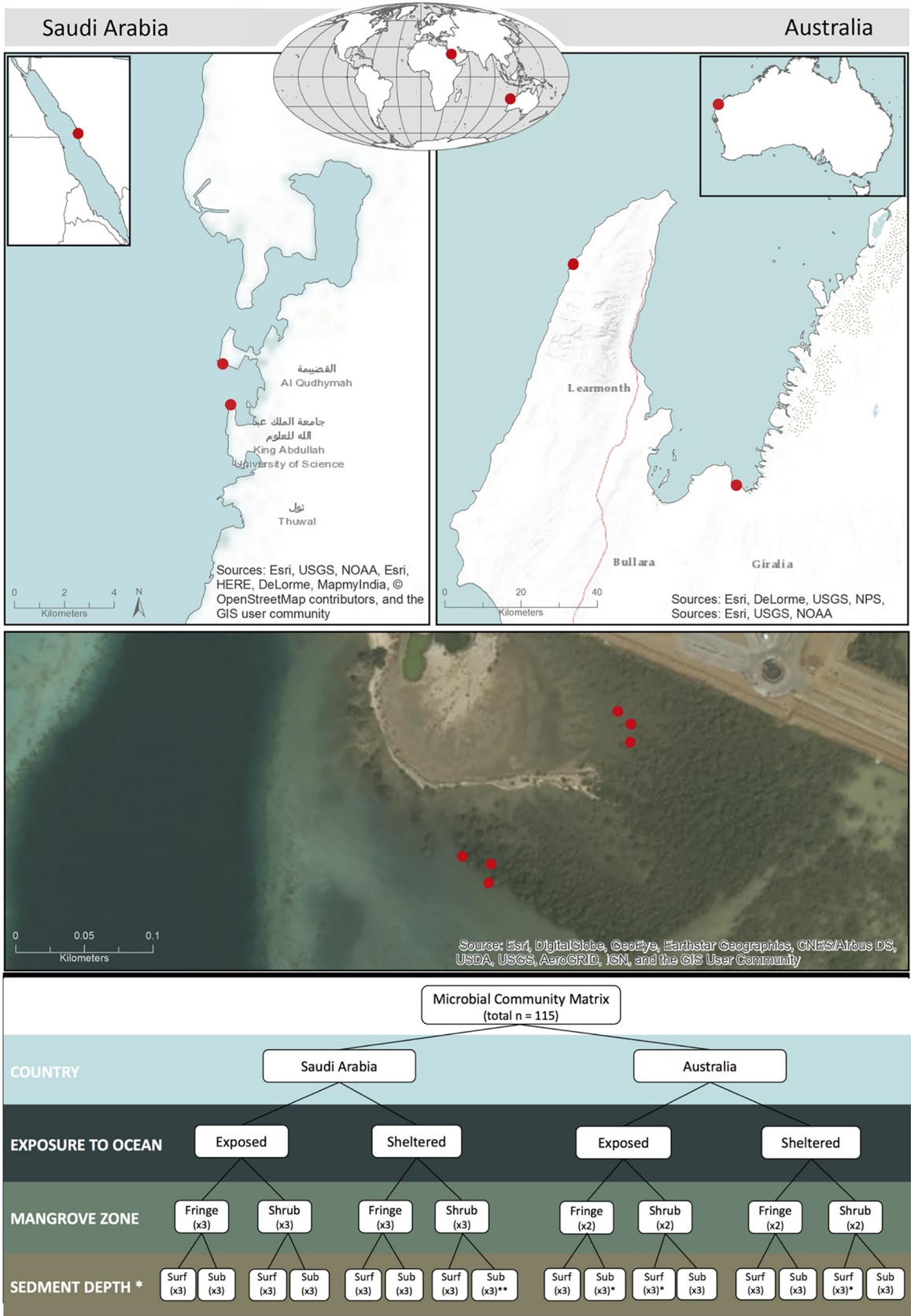

**FIG 6** Map and sampling design of the nested factors. World map showing the global regions of the sampling sites on the Western Australian coast and in Saudi Arabia. The more detailed maps show the exposed and the sheltered sites in both geographic regions. An

each sample (117). The resulting ASVs were used for taxonomic classification using the SILVA database *v. 132* (118). Eukaryotic and archaeal amplification artifacts that were also targeted by the relatively unspecific range of the V3-V4 primers used, were removed from the data set for downstream analysis. Rare ASVs (prevalence threshold of <0.5% relative to the total abundance of ASVs) were removed. Analysis of the "clean practices" negative controls identified two potentially contaminated samples. These samples showed significantly higher numbers of *Pluralibacter* and *Pseudomonas* (which are both common sources of cross-contamination in research laboratories) compared to the other samples but were still below 5% and 0.5% prevalence of all sequences of either sample, respectively. The samples were therefore omitted from further analysis. Samples that contained < 5000 reads were removed, and the remaining samples were rarefied to the minimum depth of 6048 sequences per sample (R package phyloseq *v. 1.26.1*, seed = 33) (119). In total, five samples (three from Australia, two from Saudi Arabia) were discarded due to insufficient sequencing depth (0, 0, 3517 sequences), and possible cross-contamination. Sequences were aligned (R package DECIPHER *v 2.12.0*) (120) for phylogenetic tree construction (R package phangorn *v 2.5.5*) (121), from which the alpha diversity metric of phylogenetic diversity (Faith's PD) was calculated (R package picante *v 1.8.2*) (122). From an initial number of 13,972,966 raw sequences, 10,200,365 sequences were retained after trimming and quality filtering. After merging forward and reverse reads, 4,271,706 sequences remained and after chimera removal a final 3,890,485 sequences remained. After filtration of erroneous sequences (archaeal, chloroplast, or mitochondrial sequences) and removal of rare ASVs (5% prevalence threshold), the abundance table contained 3,657 ASVs across 115 samples (Table S1). The asymptotic rarefaction curves of each sample show that the majority of the taxonomic diversity was covered sufficiently by the sequencing depth and suggested a rarefaction threshold of 5,000 sequences per sample (Fig. S1 and S2). Agglomeration of samples to the phylum and genus level resulted in 37 unique phyla and 233 genera (Table S2).

**Network analysis.** To identify ecological co-occurrence patterns in the bacterial communities, a co-occurrence network was generated for each combination of geographic region, exposure, zone, and depth using the CoNet plugin of Cytoscape 3.4 (123–125) and Gephi 0.9.1 (126) for computation and visualization, respectively. We focused on the phylotypes present in each distinct location to identify co-occurring taxa within networks that could possibly indicate functional/physical interactions in the environment (49). A combination of the Bray-Curtis (BC) and Kullback-Leiber (KLD) dissimilarity indices, along with the Pearson and Spearman correlation coefficients were used to build the networks. Edge-specific permutation and bootstrap score distributions with 1,000 iterations were performed. For each measure and edge, 100 permutations and bootstrap scores were generated. The resulting data were normalized to detect statistically significant non-random events of co-occurrences (i.e., co-presences and mutual exclusions). The *P values* were computed by z-scoring the permuted null and bootstrap confidence intervals using pooled variance (127). The most important statistical network descriptors that describe the overall structure of the network (Clustering coefficient, Centralization, Diameter, Average Neighbors, Density) were calculated using the Network Analyzer in Cytoscape (128) and the number of modules (resolution 2.5) were determined using Gephi 0.9.1.

The bacterial ASVs were represented as nodes (the central elements in the networks), from which most network parameters are calculated. Edges meanwhile represent the connections or links between the nodes, which can co-occur or be mutually exclusive. In undirected networks, such as these, no assumptions are made as to which node facilitates the presence or absence of a neighboring node. The degree of a node denotes the number of edges connecting it to other nodes, giving it a direct value of connectedness and therefore communicative importance within the network. A module is defined as a substructure within the network that exhibits a higher level of connectivity between its members compared to other members of the same network and can be used as a measure of compartmentalization of the network. Betweenness centrality can be interpreted as a measure of relevance of a node in connecting functional modules (strongly connected groups of nodes) of bacteria, with a higher score indicating an elevated importance in facilitating such a connection. Betweenness centrality can be a stronger representation of an important connection within a network, since it favors nodes that join interconnected communities rather than single nodes. The topological coefficient is a relative measure to which extent a node shares neighbors with other nodes. It therefore shows the relative connectedness of a node within its module as opposed to the entire network. Hence, nodes with a high topological coefficient are central to their module, possibly by creating an environment suitable for their co-occurring neighbors. Network analysis can also be used to identify keystone taxa (46). These are highly connected taxa that facilitate the connectivity in the network, and thereby have an increased influence on the structure and functioning of the microbial community (44).

**Functional assignments.** The Functional Annotation of Prokaryotic taxa (FAPROTAX) database *v. 1.2.1* (110) was used to perform functional annotation of the ASVs according to literature references of known microbial metabolism. The functions were assigned to the ASV table containing taxonomic annotations (from SILVA database), using the python script "collapse_table.py" in python *v. 3.7.4* (129). A total of 1103 ASVs out of 3491 ASVs (30.7%) were assigned to 40 functions (out of 90 possible). Fifteen func-

**FIG 6** Legend (Continued)
aerial photograph visualizes the separation between the zones of each forest, with actual sampling locations at the sheltered site in Saudi Arabia separated into three plots each at the tall fringe and the shrub as an example for sampling separation within each of the zones. The schematic representation of the sampling design below, shows the arrangement of the experimental factors and the numbers of replicates taken. The values in brackets represent the number of replicates for each sample. The numbers in the last row denote the sampling depth of the surface 0–2 cm (Surf) and the subsurface 2–10 cm (Sub). Asterisks (*) represents from which site a sample that has been removed for insufficient sequencing depth or cross-contamination, making the total number of samples *n* = 115.

tions were removed due to redundancy ("aerobic chemoheterotrophy" was removed in favor of "chemoheterotrophy") if they aligned collinearly (>80%), or low prevalence (if appeared in three or less samples), resulting in a set of 25 assigned functions.

**Statistical analysis.** All statistical analyses were performed in R *v. 3.6.1* (130), using the phyloseq *v. 1.26.1* (119) and vegan *v. 2.5.6* (131) packages and all graphs were created using ggplot2 *v. 3.2.1* (132). Our explanatory variable structure consists in four categorical variables: "geographic region" (fixed and orthogonal; 2 levels: Australia and Saudi Arabia); "exposure" (fixed and orthogonal; 2 levels: exposed and sheltered); "zone" (fixed and orthogonal; 2 levels: fringe and shrub); "depth" (fixed and orthogonal; 2 levels: surface and subsurface). Two alpha diversity metrics, i.e., observed number of ASVs and Shannon diversity index, were calculated with phyloseq and differences between factors were tested using analysis of variance (ANOVA) after validating the normal distribution of residuals and homoscedasticity. Differences in Faith's PD between experimental factors were tested using ANOVA. Variance partitioning of the abundance matrix (log + 1 transformed) with respect to the factors was performed with the varpart() function in "vegan" and its significance was tested by redundancy analysis (RDA). Beta-diversity at two taxonomic levels (ASV, genus) was visualized by nonmetric multidimensional scaling (NMDS), using a Bray-Curtis dissimilarity matrix estimated from the square-root-transformed and Wisconsin double-standardized ASV table. Canonical Analysis of Principal components (CAP) was performed in Primer v6 (133). A permutational multivariate analysis of variance (PERMANOVA; permutations = 999, distance = Bray-Curtis) was performed on the log transformed abundance matrix after checking the homogeneity of multivariate dispersion, to test the difference of bacterial community composition among the factors of our experimental design. PERMANOVA (permutations = 999, distance = Bray-Curtis) was conducted to test for differences of functional assignments between factors according to the design presented above.

**Data availability.** The raw sequence data were deposited in the SRA of the NCBI (https://www.ncbi.nlm.nih.gov/sra) under accession number PRJNA720541.

## SUPPLEMENTAL MATERIAL

Supplemental material is available online only.
**SUPPLEMENTAL FILE 1**, PDF file, 4.9 MB.

## ACKNOWLEDGMENTS

We thank Eleonora Fossile for her assistance during the DNA extractions, Ute Langner for creation of the map, Jonas Thomson for the graphical assembly of the figure panels, and the editor and the anonymous reviewers whose comments have resulted in a much-improved manuscript.

T.T., M.F., S.C., C.E.L., and J.I.E. contributed to the conceptualization and design of the study. T.T., M.B.S., N.P., E.A., and J.I.E. conducted field work and laboratory analyses. T.T., M.F., E.A., and J.I.E. analyzed the data. C.E.L. and B.H.J. supported the project financially and supervised it. T.T., M.F., J.I.E. wrote the manuscript and all authors read and approved the final version of it.

This work was funded by KAUST baseline funding to B.H.J. as well as baseline funding from University of Queensland to C.E.L. A travel grant from the Red Sea Research Centre (RSRC) at KAUST was awarded to T.T. This research received no specific grant from any funding agency in the public, commercial, or nonprofit sectors.

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
