## [Reviewer comments · Microbiology Spectrum]

Microbiology Spectrum

Contrasting effects of local environmental and biogeographic factors on the composition and structure of bacterial communities in arid monospecific mangrove soils

Timothy Thomson, Marco Fusi, Morgan Bennett-Smith, Natalie Prinz, Eva Aylagas, Susanna Carvalho, Catherine Lovelock, Burton Jones, and Joanne Ellis

Corresponding Author(s): Timothy Thomson, University of Waikato

Review Timeline:

Submission Date:	July 13, 2021
Editorial Decision:	September 5, 2021
Revision Received:	September 24, 2021
Editorial Decision:	November 1, 2021
Revision Received:	November 21, 2021
Accepted:	December 11, 2021

Editor: Allison Veach

Reviewer(s): Disclosure of reviewer identity is with reference to reviewer comments included in decision letter(s). The following individuals involved in review of your submission have agreed to reveal their identity: Christiane Hassenrück (Reviewer #1); Tallita C.L. Tavares (Reviewer #2)

Transaction Report:

DOI: <https://doi.org/10.1128/Spectrum.00903-21>

September 5, 2021

Mr. Timothy Thomson
University of Waikato
Tauranga
New Zealand

Re: Spectrum00903-21 (Contrasting effects of local environmental and biogeographic factors on the composition and structure of bacterial communities in arid monospecific mangrove soils)

Dear Mr. Timothy Thomson:

Two reviewers have provided feedback regarding your work submitted to Microbiology Spectrum. Both have deemed this work of sound quality and a contribution to microbial community ecology in a biogeographic context. However, both recommend some major changes to improve the readability of the manuscript. Please take special note of one reviewer's tracked-changes provided in a pdf attachment and please include a response to their documented comments in your response.

Additionally, can the authors please explain how they define structure versus composition in this study. Depending on the study system or ecological question being tested, these terms may be synonymous. It seems as though the authors define structure differently and refer to the "structure" as network topology characteristics and composition as other aspects of community alpha and beta diversity.

Thank you for submitting your manuscript to Microbiology Spectrum. When submitting the revised version of your paper, please provide (1) point-by-point responses to the issues raised by the reviewers as file type "Response to Reviewers," not in your cover letter, and (2) a PDF file that indicates the changes from the original submission (by highlighting or underlining the changes) as file type "Marked Up Manuscript - For Review Only". Please use this link to submit your revised manuscript - we strongly recommend that you submit your paper within the next 60 days or reach out to me. Detailed information on submitting your revised paper are below.

Link Not Available

Sincerely,

Allison Veach

Journals Department
Reviewer comments:

Reviewer #1 (Comments for the Author):

It has been a pleasure for me to review this manuscript. The authors present a very well designed study to investigate the effects of local environmental and biogeographic factors on mangrove soil microbial communities. The methodology is well described, their approach scientifically sound, and the conclusions well supported by the data.

I only have a few minor comments and suggestions:

Throughout the manuscript, the authors are referring to 'abundance' of microbes and their functions. However, as their conclusions are only based on compositional sequence data, I strongly recommend to rephrase and use the term 'proportion' instead. Or clearly define at the beginning that any statements about abundance are only referring to sequence proportions.

For the interpretation of the network analysis, I would like to suggest more caution in equating network topology with ecological importance. Co-occurrence networks may rather be used to develop hypotheses about ecological interactions, than to actually prove those interactions.

L168: Is this 'formula' referring to the interaction term? If yes, I would not describe an interaction as 'consistent effect', but rather use a similar description as provided in L153.

L194: Most likely a typographical error, but what is the a) after the inner parentheses referring to?

L602: How does the 30% of ASVs used for the functional prediction translate to total sequence proportions? Do you think that the functional assignment may be biased due to differing proportions of uncharacterized taxa (not part of the FAPROTAX table) in different samples?

L603: What was your reason for excluding collinear functions from the analysis?

Submitted sequence data: As mentioned in the separate field above, I suggest to modify the metadata for the following parameter:

- library_source should be METAGENOMIC
- environmental_sample should be TRUE
- target_gene is missing and should be 16S rRNA

Reviewer #2 (Comments for the Author):

The manuscript by Thomson et al. presents the bacterial community associated with the soil under *Avicennia marina* in distinct geographical areas and in subareas of local importance (sheltered/exposed; fringe/shrub). The question is relevant and clearly exposed. The use of molecular and bioinformatics tools allowed to have a satisfactory description of the studied communities. However, the absence of physical and chemical data or other environmental data, such as temperature, rainfall, and evapotranspiration (mostly important for arid mangroves) is an important flaw. I consider this paper will be helpful however there are some concerns to address. The data analysis is robust but the authors should try to organize results and discussion to improve clarity. I have also recommended including some discussion topics for better contextualization.

Staff Comments:

Preparing Revision Guidelines

Please return the manuscript within 60 days; if you cannot complete the modification within this time period, please contact me. If you do not wish to modify the manuscript and prefer to submit it to another journal, please notify me of your decision immediately so that the manuscript may be formally withdrawn from consideration by Microbiology Spectrum.

**Contrasting effects of local environmental and biogeographic**
**factors on the composition and structure of bacterial communities**
**in arid monospecific mangrove soils**

Thomson, T.^{1,2,*}, Fusi, M.^{2,3}, Bennett-Smith, M. F.², Prinz, N.¹, Aylagas, E.², Carvalho, S.², Lovelock,
C. E.⁴, Jones, B. H.², Ellis, J. I.^{1,2}

University of Waikato, School of Sciences, 101 Durham Street, Tauranga, New Zealand 3110

King Abdullah University of Science and Technology (KAUST), Biological and Environmental
Sciences and Engineering Division (BESE), Thuwal, 23955-6900, Saudi Arabia

School of Applied Sciences, Edinburgh Napier University, Edinburgh, UK

School of Biological Sciences, The University of Queensland, St Lucida, QLD 4072, Australia

Corresponding author email: timi.thomson@gmail.com

OrcidID:

TT: 0000-0001-6618-7080

MF: 0000-0001-7433-2487

MBS: 0000-0002-7681-8939

NP: 0000-0002-3543-4242

SC: 0000-0003-1300-1953

BJ: 0000-0002-9599-1593

EA: 0000-0001-9792-8451

CEL: 0000-0002-2219-6855

JIE: 0000-0002-2625-4274

Abstract

[revised manuscript text omitted]

Two contrasting theories discuss the nature of microbial distributions: one describes the
microbiome as a fluent continuum across the globe, which is formed by its environmental
settings and shows the same community composition when the conditions are similar, i.e.
“everything is everywhere *but* the environment selects”^{32–35}. The second hypothesis states
that microorganisms follow biogeographic patterns similar to those of macro-organisms,
that are limited by a combination of historical and location settings (selection, drift,
dispersal, and mutation)^{36–40}. As soil microbial communities of mangrove forests facilitate
ecosystem processes of global importance, understanding the mechanisms that shape
microbial community composition can aid in predicting changes resulting from fluctuating
environmental and climatic conditions^{12,36}. This study is the first to our knowledge that
assesses the variation in bacterial communities in mangrove forests across varying spatial
scales, to provide insight into the global distribution of bacterial communities in this unique
habitat.

Recent work indicates that, in order to assure the functioning of an ecosystem, the structure
of its microbial community is as important as its composition^{41–44}. Network analyses of co-
occurring microbial taxa have proven useful in the characterization of microbial community
structures, despite the limitations of this statistical method, providing a solid sampling
design and sufficient replication is considered^{43,45,46}. Disentangling patterns of
connectedness between members of the microbial community and their spatial variability is
an important step toward understanding the importance of single members or groups of the
microbial community in facilitating functional diversity and resilience⁴⁷. The functional
diversity of a system describes the traits of organisms present within it that potentially
influence or contribute to changes in this system⁴⁸. Bacterial communities can be

categorized into functional groups such as nitrifying and denitrifying bacteria, nitrogen fixing
bacteria, phototrophic bacteria, and sulfate-oxidizing bacteria^{5,8,19}. Ecological and functional
microbial redundancies are key drivers of ecosystem resilience⁴⁶. As mangroves are
threatened by numerous factors (e.g., sea level rise, over-exploitation, drought,
increased/decreased salinity)⁴⁹⁻⁵¹, patterns of microbial biodiversity can provide insights on
the effective capacity of the mangrove to buffer these changes.

Based on the evidence above, we hypothesized that local environmental parameters are
important controlling factors of the soil bacterial community, which may exceed the
importance of geographic region. Therefore, we expected soil bacterial communities
associated with the same species of mangroves and under similar climate conditions to vary
more on local scales compared to variation at global scales. Moreover, physico-chemical
conditions in different forests and between forest zones will have a significant influence on
soil bacterial networks and the keystone taxa within them.

To test our hypotheses, we compared the bacterial community of two monospecific
*Avicennia marina* mangrove forests over varying spatial scales using 16S rRNA gene
sequencing, aiming to identify the selective forces that determine the bacterial assembly
patterns within the microbiome in this highly specialised environment. We analyzed the
bacterial communities at different depths in the soil, distance from the sea, in forests with
different exposures to oceanic influences, and from two distant geographic regions that are
both dominated by a hot and arid climate (i.e. Saudi Arabia and Australia)⁵².

**Results**

Diversity of bacterial communities

The rank-abundance relationship displayed in Figure 1A shows the disproportional
abundance of rare ASVs (high rank values) compared to few highly abundant ASVs in the
community, which is consistent across all sampling sites. Species richness generally showed
significant differences in number of ASVs between Australia and Saudi Arabia (Figure 1B,
Supplementary table S4). In Australia, species richness was similar between the zones of the
exposed site, while it varied significantly between zones of the sheltered site. Here the
richness was higher in the soils of the tall fringing forest. In Saudi Arabia, species richness
was generally higher in the shrub than in the fringe, although only significant for the surface
samples (Figure 1B, Supplementary table S4). The variation of species richness between
depths was significant in the fringe of both exposures in Australia and in the shrub of both
exposures in Saudi Arabia, with the richness being higher in the surface samples except for
the sheltered site in Australia. The species richness differed significantly for each
combination of geographic region, exposure, and zone but not for depth (geographic region
154 \times exposure \times zone; $F_{1,103} = 8.83$, p-value = 0.004).

Except for the exposed site in Australia, the differences between depths in the soil were
significant in the shrub zone of the forest, with higher diversity scores in the surface layers
of the soil (Figure 1C). The variation of the Shannon diversity index was only significant
between the interaction of local factors (zone \times depth; $F_{1,104} = 5.34$, p-value 0.02;
Supplementary table S6).

Phylogenetic diversity was markedly higher in the shrub of Saudi Arabia compared to all
other sites, while differences between zones in Australia were inconsistent (Figure 1D). As

with the Species richness and the Shannon diversity index, the phylogenetic diversity of the
sheltered site in Australia showed the opposite trend between the zones, yielding higher
diversity in the fringe compared to the shrub. The differences in phylogenetic diversity are
consistent between the factors geographic region, exposure, and zone (geographic region ×
exposure × zone; $F_{1,103} = 14.66$, p -value < 0.001; Supplementary table S8). A table of the
diversity metrics for each sample, and the post-hoc test results are displayed in the
supplementary information (Supplementary tables S3, S5, and S7).

*Proteobacteria* were the most abundant phylum across all samples (58.96%), followed by
*Bacteroidetes* (17.44%), *Chloroflexi* (8.06%), *Calditrichaeota* (3.81%), and *Nitrospirae* (1.59%),
contributing to 90% of the total community. Within *Proteobacteria*, the most abundant class
was *Deltaproteobacteria* (31.60%), followed by *Gammaproteobacteria* (23.34%), *Bacteroidia*
(13.65%), *Anaerolineae* (7.12%), and *Alphaproteobacteria* (4.44%). The relative abundances
of phyla between factors are shown in Figure 2A. In the shrub zone, the differences in
relative abundance of phyla between surface and subsurface was more obvious than in the
tall fringing zone. This is consistent for exposed and sheltered sites in both geographic
regions. *Proteobacteria* dominated the community in every location and the relative
abundance of this phylum was higher in the subsurface layer than at the surface
(Supplementary figure S12). The second most abundant phylum, *Bacteroidetes*, was more
abundant in the surface layer of all locations (Supplementary figure S13). *Chloroflexi*,
*Calditrichaeota*, and *Nitrospirae* were more abundant in the subsurface layer, except from
the fringe samples of the sheltered site in Australia (Supplementary figures S14 to S16).
*Cyanobacteria* were dominant at the surface and more so in Saudi Arabian samples, with

the exposed shrub of Australian samples also showing slightly elevated abundances
(Supplementary figure S17).

Variance partitioning in combination with redundancy analysis (RDA) showed that 57.72% of
the total variation in the community matrix was explained by all four experimental factors
(Supplementary table S12). The variance explained by the individual factors was 7.2% by
geographic region, 3.6% by exposure, 10.0% by zone, and 3.5% by depth ($p < 0.001$, 999
permutations; see Figure 2B and Supplementary table S12 and S13), a). The cumulative
variation of the individual factors does not include variation that can be explained by a
combination of two or more factors (overlap in the venn-diagram (Supplementary figure
S18). Therefore, the residual variation between what is explained by the full model (57.72%)
and the individual factors combined (24.82%), infers the variation that can be explained by
the interactions between the factors (32.90%).

The NMDS plot (Stress = 0.18), calculated from the community matrix at ASV level,
separates the samples by the factor zone along the first axis (Figure 2C). The factor
geographic region mainly accounts for the spread of the data points along the second axis.
This is more obvious for the fringe samples, whereas the samples from the shrub overlap
slightly, leading to a slight overlap of communities from different geographic regions. The
factors exposure and depth fail to separate the samples into distinguishable groups. At
genus level (Stress = 0.19), the factor zone still separated the samples, however the factor
geographic region did not (Figure 2D).  Similar to the ASV level, the factor exposure did not

contribute to separating the data in the ordination space. The models calculated from the

CAP analysis were able to match the bacterial communities with the correct factor

combination between 80 – 100% of the time at ASV level and genus level (13 and 11 perfect
scores out of 16, respectively). The model at phylum level performed poorly (between 44.4
213 – 100%), which is why no further analysis was carried out at phylum level.

PERMANOVA showed a significant interaction on the taxonomic community composition at
ASV level across all factors (geographic region × exposure × zone × depth; $F_{1,114} = 3.36$; p-value
= 0.001). A pairwise comparison of the significant four-way interaction term confirmed the
statistical significance between ecologically relevant combinations of the factors geographic
region, exposure, zone, and depth (Supplementary table S20). A significant interaction term
between geographic region, exposure, zone, and depth was found at genus level
(geographic region × exposure × zone × depth; $F_{1,114} = 3.99$; p-value 0.002; Supplementary
table S21).

**Network analysis**

The network analysis revealed distinct co-occurrence patterns between bacterial ASVs of
mangrove soils across the factors of this study. The Australian networks contained a lower
average number of nodes and a higher number of edges than Saudi Arabia, resulting in a
higher density and average number of neighbors, but lower network diameter. The
modularity, as indicated by the number of modules detected in each network, was higher in
Saudi Arabia, whereas the clustering coefficient and the centralisation was higher in
Australian networks (Table 1, Figure 3).

The betweenness centrality metric per phylum showed differences between the
experimental factors (Figure 4A). *Cyanobacteria* showed the highest betweenness values in
the surface samples, especially in Australian soils. Exceptions were the exposed fringe
samples of both geographic regions, with a low and a high betweenness score of
*Cyanobacteria* in Australia and Saudi Arabia, respectively. *Gemmatimodetes* also showed
high betweenness values within most samples. Betweenness values for *Proteobacteria* were
low in all but one of the samples from Saudi Arabia and seemed to be overall more
important in Australian networks. *Acidobacteria* showed higher betweenness in subsurface
samples, and *Actinobacteria* seemed to be important in some samples but no pattern was
discernible.

The topological coefficient additionally highlighted the role of different phyla (Figure 4B).
Overall, *Spirochaetes*, *Zixibacteria*, and in some samples *Gemmatimodetes* had the highest
scores. *Spirochaetes* were more prominent in subsurface soils, except for the exposed fringe
samples in both geographic regions. *Zixibacteria* had slightly higher values in the fringe
samples of both geographic regions, but showed higher presence in sheltered forests in
Australia, opposed to higher presence in exposed forests in Saudi Arabia. *Gemmatimodetes*
showed high values in the soils of sheltered sites of Australian forests. Keystone taxa
analysis showed an increased number of important nodes in Saudi Arabian as compared to
Australian networks (Supplementary figure S27).

In Australian networks, the relative proportion of nodes at higher degrees (normalized) was
dominated by single phyla, especially in the shrub forests (Figure 5). In the surface samples

[revised manuscript text omitted]

Sampling was conducted in June 2018 in Australia and in December 2018 in Saudi Arabia,
hence during the winter season in both regions. **A nested design was chosen to sample the**
**tall fringe and dwarfed shrub zones of the mangrove forest at the exposed and sheltered**

sites in both geographic regions. Six cores per zone were sampled for the bacterial
community composition in Australia using a 3 cm diameter core, extending approximately
10 cm into the soil. In Saudi Arabia, nine cores were sampled per zone (as triplicates of each
nested location; see Figure 7). In the field, all cores were subsequently separated into
surface (0 – 2 cm) and subsurface (5 – 7 cm) fractions. The soil depths at the exposed site in
Saudi Arabia were very shallow (max. 5 cm), due to an underlying hard carbonate platform.
In these environments, the deepest possible depth was sampled (3 – 5 cm). All samples
were kept in the dark on ice and taken to the laboratory for storage at -80 °C, immediately
after the sampling was completed (2 - 3 hours). Across geographic regions (Saudi
Arabia/Australia), coastal exposures (exposed/sheltered), forest zones (fringe/shrub), and
soil depths (surface/subsurface), a total of 120 samples were collected.

**DNA extraction and target gene amplification**

DNA for bacterial community analysis was extracted from approximately 0.8 g of soil, using
the DNeasy[®]PowerSoil[®] Kit (Qiagen, Hilden, Germany). Using the specific primers 341F and
805R¹⁰³ with overhang illumina adapters (Illumina Inc., San Diego, CA, USA), the V3-V4
hypervariable regions of the 16S rRNA gene were amplified by polymerase chain reaction
(PCR), at a final reaction volume of 25 µL per sample¹⁰⁴. The amplified samples were
cleaned to remove primer-dimers and non-targeted DNA molecules using the SequalPrep[®]
Normalization Plate (96) Kit (Invitrogen, Carlsbad, CA, USA). Library preparation was carried
out using the Nextera[®] XT Index kit (Illumina Inc., San Diego, CA, USA) in combination with
the Qiagen Multiplex PCR Master Mix (Qiagen, Hilden, Germany) that uses the HotStarTaq
DNA Polymerase. The cycler was set to 95 °C as initial temperature for 15 minutes, followed
by 8 cycles of 95 °C for 30 seconds, 55 °C for 90 seconds, and 72 °C for 30 seconds, and a

[revised manuscript text omitted]

MBS, NP, EA, and JIE conducted field work and laboratory analyses. TT, MF, EA, and JIE
analyzed the data. CEL and BHJ supported the project financially and supervised it. TT, MF,
JIE wrote the manuscript and all authors read and approved the final version of it.

**Competing interests**

The authors declare that they have no competing interests.

**Funding**

This work was funded by KAUST baseline funding to BHJ as well as baseline funding from
University of Queensland to CEL. A travel grant from the Red Sea Research Centre (RSRC) at
**KAUST** was awarded to TT. This research received no specific grant from any funding agency
in the public, commercial, or non-profit sectors.

**Supplementary material**

Supplementary File 1: Supplementary information workflow. RMarkdown of the data
analysis including supplementary figures and tables as indicated throughout the manuscript
(PDF; 2.9 MB)

**References**

- 1. Donato, D. C. *et al.* Mangroves among the most carbon-rich forests in the tropics. *Nat. Geosci.* **4**, 293–
297 (2011).
- 2. Alongi, D. M. Global Significance of Mangrove Blue Carbon in Climate Change Mitigation. *Sci* **2**, 67
(2020).
- 3. Alongi, D. M. The role of bacteria in nutrient recycling in tropical mangrove and other coastal benthic
ecosystems. *Hydrobiologia* **285**, 19–32 (1994).
- 4. Alongi, D. M. Bacterial productivity and microbial biomass in tropical mangrove sediments. *Microb.*
*Ecol.* **15**, 59–79 (1988).
- 5. Holguin, G., Vazquez, P. & Bashan, Y. The role of sediment microorganisms in the productivity,
conservation, and rehabilitation of mangrove ecosystems: an overview. *Biol. Fertil. Soils* **33**, 265–278
(2001).
- 6. Soldan, R. *et al.* Bacterial endophytes of mangrove propagules elicit early establishment of the natural
host and promote growth of cereal crops under salt stress. *Microbiol. Res.* (2019)
doi:10.1016/J.MICRES.2019.03.008.
- 7. Alongi, D. M. Mangrove-Microbe-Soil Relations. 85–103 (2005).
- 8. Reef, R., Feller, I. C. & Lovelock, C. E. Nutrition of mangroves. *Tree Physiol.* **30**, 1148–1160 (2010).
- 9. Trevathan-Tackett, S. M. *et al.* A horizon scan of priorities for coastal marine microbiome research. *Nat.*
*Ecol. Evol.* **3**, 1509–1520 (2019).
- 10. Thompson, L. R. *et al.* A communal catalogue reveals Earth’s multiscale microbial diversity. *Nature* **551**,
457–463 (2017).
- 11. Bahram, M. *et al.* Structure and function of the global topsoil microbiome. *Nature* **560**, 233–237 (2018).
- 12. Cavicchioli, R. *et al.* Scientists’ warning to humanity: microorganisms and climate change. *Nat. Rev.*
*Microbiol.* (2019) doi:10.1038/s41579-019-0222-5.
- 13. Ferreira, T. O. *et al.* Spatial patterns of soil attributes and components in a mangrove system in
Southeast Brazil (São Paulo). *J. Soils Sediments* **10**, 995–1006 (2010).
- 14. Odum, W. E. & Heald, E. J. Mangrove Forests and Aquatic Productivity. in *Coupling of land and water*
*systems* 129–136 (Springer, Berlin, Heidelberg, 1975). doi:10.1007/978-3-642-86011-9_5.
- 15. Odum, W. E. & Heald, E. J. The detritus-based food web of an estuarine mangrove community. in

- *Estuarine research. Volume I, Chemistry, biology, and the estuarine system* 738 (Academic Press, Inc,
1975).
- 16. Moitinho, M. A., Bononi, L., Souza, D. T., Melo, I. S. & Taketani, R. G. Bacterial Succession Decreases
Network Complexity During Plant Material Decomposition in Mangroves. *Microb. Ecol.* **76**, 954–963
(2018).
- 17. Clark, M. W., McConchie, D., Lewis, D. W. & Saenger, P. Redox stratification and heavy metal
partitioning in Avicennia-dominated mangrove sediments: a geochemical model. *Chem. Geol.* **149**,
147–171 (1998).
- 18. Penha-Lopes, G. *et al.* The role of biogenic structures on the biogeochemical functioning of mangrove
constructed wetlands sediments – A mesocosm approach. *Mar. Pollut. Bull.* **60**, 560–572 (2010).
- 19. Booth, J. M., Fusi, M., Marasco, R., Mbobo, T. & Daffonchio, D. Fiddler crab bioturbation determines
consistent changes in bacterial communities across contrasting environmental conditions. *Sci. Rep.* **9**,
3749 (2019).
- 20. Kuramae, E. E. *et al.* Soil characteristics more strongly influence soil bacterial communities than land-
use type. *FEMS Microbiol. Ecol.* **79**, 12–24 (2012).
- 21. Feller, I. C., Mckee, K. L., Whigham, D. F. & O’neill, J. P. Nitrogen vs. Phosphorus Limitation across an
Ecotonal Gradient in a Mangrove Forest. *Biogeochemistry* **62**, 145–175 (2003).
- 22. Lovelock, C. E., Feller, I. C., Mckee, K. L. & Thompson, R. Variation in Mangrove Forest Structure and
Sediment Characteristics in Bocas del Toro, Panama. *Caribbean J. Sci.* **41**, 456–464 (2005).
- 23. Holguin, G. *et al.* Mangrove health in an arid environment encroached by urban development—a case
study. *Sci. Total Environ.* **363**, 260–274 (2006).
- 24. Feller, I. C., Lovelock, C. E. & McKee, K. L. Nutrient Addition Differentially Affects Ecological Processes
of Avicennia germinans in Nitrogen versus Phosphorus Limited Mangrove Ecosystems. *Ecosystems* **10**,
347–359 (2007).
- 25. Feller, I. C. *et al.* Biocomplexity in Mangrove Ecosystems. *Annu. Rev. Mar. Sci.* **2**, 395–417 (2010).
- 26. Gonzalez-Acosta, B., Bashan, Y., Hernandez-Saavedra, N. Y., Ascencio, F. & Cruz-Agüero, G. Seasonal
seawater temperature as the major determinant for populations of culturable bacteria in the
sediments of an intact mangrove in an arid region. *FEMS Microbiol. Ecol.* **55**, 311–321 (2006).
- 27. Gomes, N. C. M. *et al.* Mangrove microniches determine the structural and functional diversity of
enriched petroleum hydrocarbon-degrading consortia. *FEMS Microbiol. Ecol.* **74**, 276–290 (2010).
- 28. Vera-Gargallo, B. *et al.* Spatial distribution of prokaryotic communities in hypersaline soils. *Sci. Rep.* **9**,
1769 (2019).
- 29. Andreote, F. D. *et al.* The Microbiome of Brazilian Mangrove Sediments as Revealed by Metagenomics.
*PLoS One* **7**, e38600 (2012).
- 30. Imchen, M. *et al.* Searching for signatures across microbial communities: Metagenomic analysis of soil
samples from mangrove and other ecosystems. *Sci. Rep.* **7**, 8859 (2017).
- 31. Imchen, M. *et al.* Comparative mangrove metagenome reveals global prevalence of heavy metals and
antibiotic resistome across different ecosystems. *Sci. Rep.* **8**, 11187 (2018).
- 32. Becking, L. B. *Geobiologie of inleiding tot de milieukunde.* (W.P. Van Stockum & Zoon, 1934).
- 33. Fenchel, T. & Finlay, B. J. The Ubiquity of Small Species : Patterns of Local and Global Diversity.
*Bioscience* **54**, (2004).
- 34. Fierer, N. & Jackson, R. B. The diversity and biogeography of soil bacterial communities. *PNAS* **103**,
626–631 (2006).
- 35. Finlay, B. J. Global dispersal of free-living microbial eukaryote species. *Science (80-)*. **296**, 1061–1063

- (2002).
- 36. Martiny, J. B. H. *et al.* Microbial biogeography : putting microorganisms on the map. *Nat. Rev.*
*Microbiol.* **4**, 102–112 (2006).
- 37. Hanson, C. A., Fuhrman, J. A., Horner-devine, M. C. & Martiny, J. B. H. Beyond biogeographic patterns:
processes shaping the microbial landscape. *Nat. Rev. Microbiol.* **10**, 497–506 (2012).
- 38. Langenheder, S. & Lindström, E. S. Factors influencing aquatic and terrestrial bacterial community
assembly. *Environ. Microbiol. Rep.* **11**, 306–315 (2019).
- 39. Vellend, M. Conceptual synthesis in community ecology. *Q. Rev. Biol.* **85**, 183–206 (2010).
- 40. Nemergut, D. R. *et al.* Patterns and Processes of Microbial Community Assembly. *Microbiol. Mol. Biol.*
*Rev.* **77**, 342–356 (2013).
- 41. Banerjee, S., Schlaeppli, K. & van der Heijden, M. G. A. Keystone taxa as drivers of microbiome structure
and functioning. *Nat. Rev. Microbiol.* **16**, 567–576 (2018).
- 42. Banerjee, S. *et al.* Agricultural intensification reduces microbial network complexity and the abundance
of keystone taxa in roots. *ISME J.* **13**, 1722–1736 (2019).
- 43. Berry, D. & Widder, S. Deciphering microbial interactions and detecting keystone species with co-
occurrence networks. *Front. Microbiol.* **5**, 219 (2014).
- 44. Layeghifard, M., Hwang, D. M. & Guttman, D. S. Disentangling Interactions in the Microbiome: A
Network Perspective. *Trends Microbiol.* **25**, 217–228 (2017).
- 45. Heleno, R., Devoto, M. & Pocock, M. Connectance of species interaction networks and conservation
value: Is it any good to be well connected? *Ecol. Indic.* **14**, 7–10 (2012).
- 46. Fuhrman, J. A. Microbial community structure and its functional implications. *Nature* **459**, (2009).
- 47. Pearman, J. K. *et al.* Disentangling the complex microbial community of coral reefs using standardized
Autonomous Reef Monitoring Structures (ARMS). *Mol. Ecol.* **28**, 3496–3507 (2019).
- 48. Tilman, D. Functional Diversity. *Encycl. Biodivers.* 109–120 (2001) doi:10.1016/B0-12-226865-2/00132-
2.
- 49. Duke, N. C. *et al.* A World Without Mangroves ? *Science (80-.)*. **317**, 41–43 (2007).
- 50. Polidoro, B. A. *et al.* The Loss of Species: Mangrove Extinction Risk and Geographic Areas of Global
Concern. *PLoS One* **5**, e10095 (2010).
- 51. Saintilan, N. *et al.* Thresholds of mangrove survival under rapid sea level rise. *Science* **368**, 1118–1121
(2020).
- 52. Adame, M. F. *et al.* Mangroves in arid regions: Ecology, threats, and opportunities. *Estuar. Coast. Shelf*
*Sci.* (2020) doi:10.1016/j.ecss.2020.106796.
- 53. Leopold, A., Marchand, C., Deborde, J., Chaduteau, C. & Allenbach, M. Influence of mangrove zonation
on CO2 fluxes at the sediment-air interface (New Caledonia). *Geoderma* **202–203**, 62–70 (2013).
- 54. Hugerth, L. W. *et al.* Metagenome-assembled genomes uncover a global brackish microbiome. *Genome*
*Biol.* **16**, 279 (2015).
- 55. Sunagawa, S. *et al.* Ocean plankton. Structure and function of the global ocean microbiome. *Science*
**348**, 1261359 (2015).
- 56. Rocha, L. L., Colares, G. B., Nogueira, V. L. R., Paes, F. A. & Melo, V. M. M. Distinct Habitats Select
Particular Bacterial Communities in Mangrove Sediments. *Int. J. Microbiol.* **2016**, (2016).
- 57. Fodelianakis, S. *et al.* Dispersal homogenizes communities via immigration even at low rates in a
simplified synthetic bacterial metacommunity. *Nat. Commun.* **10**, 23955–6900 (2019).

- 58. Jiang, X.-T. *et al.* Illumina Sequencing of 16S rRNA Tag Revealed Spatial Variations of Bacterial
Communities in a Mangrove Wetland. *Microb. Ecol.* **66**, 96–104 (2013).
- 59. Thrush, S. F., Gray, J. S., Hewitt, J. E. & Ugland, K. I. *Predicting the Effects of Habitat Homogenization on*
*Marine Biodiversity*. vol. 16 (2006).
- 60. Hewitt, J., Thrush, S., Lohrer, A. & Townsend, M. A latent threat to biodiversity: Consequences of small-
scale heterogeneity loss. *Biodivers. Conserv.* **19**, 1315–1323 (2010).
- 61. Vellend, M. *et al.* Homogenization of forest plant communities and weakening of species-environment
relationships via agricultural land use. *J. Ecol.* **95**, 565–573 (2007).
- 62. Fierer, N., Schimel, J. P. & Holden, P. A. Variations in microbial community composition through two
soil depth profiles. *Soil Biol. Biochem.* **35**, 167–176 (2003).
- 63. Montgomery, J. M., Bryan, K. R., Horstman, E. M. & Mullarney, J. C. Attenuation of tides and surges by
mangroves: Contrasting case studies from New Zealand. *Water (Switzerland)* **10**, (2018).
- 64. Lindström, E. S. & Langenheder, S. Local and regional factors influencing bacterial community assembly.
*Environ. Microbiol. Rep.* **4**, 1–9 (2012).
- 65. Bay, S. K. *et al.* Soil Bacterial Communities Exhibit Strong Biogeographic Patterns at Fine Taxonomic
Resolution. *mSystems* **5**, (2020).
- 66. Green, J. & Bohannan, B. J. M. Spatial scaling of microbial biodiversity. *Trends Ecol. Evol.* **21**, (2006).
- 67. Ramette, A. & Tiedje, J. M. Biogeography: An Emerging Cornerstone of Understanding Prokaryotic
Diversity, Ecology, and Evolution. *Microb. Ecol.* **53**, 197–207 (2007).
- 68. Acker, J., Leptoukh, G., Shen, S., Zhu, T. & Kempler, S. Remotely-sensed chlorophyll a observations of
the northern Red Sea indicate seasonal variability and influence of coastal reefs. *J. Mar. Syst.* **69**, 191–
204 (2008).
- 69. Carvalho, S., Kürten, B., Krokos, G., Hoteit, I. & Ellis, J. I. The Red Sea. *World Seas an Environ. Eval.* 49–
74 (2019) doi:10.1016/B978-0-08-100853-9.00004-X.
- 70. McLachlan, J. R., Haghkerdar, J. M. & Greig, H. S. Strong zonation of benthic communities across a tidal
freshwater height gradient. *Freshw. Biol.* **64**, 1284–1294 (2019).
- 71. Boehm, A. B., Yamahara, K. M. & Sassoubre, L. M. Diversity and transport of microorganisms in
intertidal sands of the California coast. *Appl. Environ. Microbiol.* **80**, 3943–3951 (2014).
- 72. Logares, R. *et al.* Disentangling the mechanisms shaping the surface ocean microbiota. *Microbiome* **8**,
1–17 (2020).
- 73. Stegen, J. C. *et al.* Quantifying community assembly processes and identifying features that impose
them. *ISME J.* **7**, 2069–2079 (2013).
- 74. Fulthorpe, R. R., Roesch, L. F. W., Riva, A. & Triplett, E. W. Distantly sampled soils carry few species in
common. *ISME J.* **2**, 901–910 (2008).
- 75. Power, J. F. *et al.* Microbial biogeography of 925 geothermal springs in New Zealand. *Nat. Commun.* **9**,
1–12 (2018).
- 76. Whitaker, R. J., Grogan, D. W. & Taylor, J. W. Geographic Barriers Isolate Endemic Populations of
Hyperthermophilic Archaea. *Science (80-.)*. **301**, 976–978 (2003).
- 77. Valverde, A., Makhalyane, T. P., Seely, M. & Cowan, D. A. Cyanobacteria drive community
composition and functionality in rock-soil interface communities. *Mol. Ecol.* **24**, 812–821 (2015).
- 78. Kumar, M., Singh, D. P., Prabha, R. & Sharma, A. K. Role of cyanobacteria in nutrient cycle and use
efficiency in the soil. in *Nutrient Use Efficiency: From Basics to Advance* (eds. Rakshit, A., Singh, H. B. &
Sen, A.) 163–171 (Springer, 2015). doi:10.1007/978-81-322-2169-2.

- 79. Stal, L. J. Cyanobacterial mats and stromatolites. in *Ecology of Cyanobacteria II: Their Diversity in Space*
*and Time* vol. 9789400738553 65–125 (Springer Netherlands, 2012).
- 80. Hanada, S. & Sekiguchi, Y. The phylum gemmatimonadetes. in *The Prokaryotes: Other Major Lineages*
*of Bacteria and The Archaea* 677–681 (Springer-Verlag Berlin Heidelberg, 2014). doi:10.1007/978-3-
642-38954-2_164.
- 81. Zhang, H. *et al.* Gemmatimonas aurantiaca gen. nov., sp. nov., a Gram-negative, aerobic,
polyphosphate-accumulating micro-organism, the first cultured representative of the new bacterial
phylum Gemmatimonadetes phyl. nov. *Int. J. Syst. Evol. Microbiol.* **53**, 1155–1163 (2003).
- 82. DeBruyn, J. M., Nixon, L. T., Fawaz, M. N., Johnson, A. M. & Radosevich, M. Global biogeography and
quantitative seasonal dynamics of Gemmatimonadetes in soil. *Appl. Environ. Microbiol.* **77**, 6295–6300
(2011).
- 83. Pascualt, N. *et al.* Stimulation of Different Functional Groups of Bacteria by Various Plant Residues as a
Driver of Soil Priming Effect. *Ecosystems* **16**, 810–822 (2013).
- 84. Castelle, C. J. *et al.* Extraordinary phylogenetic diversity and metabolic versatility in aquifer sediment.
*Nat. Commun.* **4**, 1–10 (2013).
- 85. Momper, L., Jungbluth, S. P., Lee, M. D. & Amend, J. P. Energy and carbon metabolisms in a deep
terrestrial subsurface fluid microbial community. *ISME J.* **11**, 2319–2333 (2017).
- 86. Strous, M. *et al.* Deciphering the evolution and metabolism of an anammox bacterium from a
community genome. *Nature* **440**, 790–794 (2006).
- 87. Zhao, R., Summers, Z. M., Christman, G. D., Yoshimura, K. M. & Biddle, J. F. Metagenomic views of
microbial dynamics influenced by hydrocarbon seepage in sediments of the Gulf of Mexico. *Sci. Rep.* **10**,
1–13 (2020).
- 88. Dong, X. *et al.* Fermentative Spirochaetes mediate necromass recycling in anoxic hydrocarbon-
contaminated habitats. *ISME J.* **12**, 2039–2050 (2018).
- 89. Dubinina, G., Grabovich, M., Leshcheva, N., Rainey, F. A. & Gavrish, E. Spirochaeta perfilievii sp. nov.,
an oxygen-tolerant, sulfide-oxidizing, sulfur- and thiosulfate-reducing spirochaete isolated from a
saline spring. *Int. J. Syst. Evol. Microbiol.* **61**, 110–117 (2011).
- 90. Delgado-Baquerizo, M. *et al.* Multiple elements of soil biodiversity drive ecosystem functions across
biomes. *Nat. Ecol. Evol.* **4**, 210–220 (2020).
- 91. Herren, C. M. & McMahon, K. D. Keystone taxa predict compositional change in microbial communities.
*Environ. Microbiol.* **20**, 2207–2217 (2018).
- 92. Jousset, A. *et al.* Where less may be more: How the rare biosphere pulls ecosystems strings. *ISME*
*Journal* vol. 11 853–862 (2017).
- 93. Lynch, M. D. J. & Neufeld, J. D. Ecology and exploration of the rare biosphere. *Nature Reviews*
*Microbiology* vol. 13 217–229 (2015).
- 94. Kristensen, E., Holmer, M. & Bussarawit, N. Benthic metabolism and sulfate reduction in a Southeast
Asian mangrove swamp. *Mar. Ecol. Prog. Ser.* **73**, 93–103 (1991).
- 95. Varon-Lopez, M. *et al.* Sulphur-oxidizing and sulphate-reducing communities in Brazilian mangrove
sediments. *Environ. Microbiol.* **16**, 845–855 (2014).
- 96. Bartlett, K. B., Bartlett, D. S., Harriss, R. C. & Sebacher, D. I. Methane emissions along a salt marsh
salinity gradient. *Biogeochemistry* **4**, 183–202 (1987).
- 97. Poffenbarger, H. J., Needelman, B. A. & Magonigal, J. P. Salinity Influence on Methane Emissions from
Tidal Marshes. *Wetlands* **31**, 831–842 (2011).
- 98. Zhang, X., Hu, B. X., Ren, H. & Zhang, J. Composition and functional diversity of microbial community

- across a mangrove-inhabited mudflat as revealed by 16S rDNA gene sequences. *Sci. Total Environ.* **633**,
518–528 (2018).
- 99. Boschker, H. T. S., Vasquez-Cardenas, D., Bolhuis, H., Moerdijk-Poortvliet, T. W. C. & Moodley, L.
Chemoautotrophic Carbon Fixation Rates and Active Bacterial Communities in Intertidal Marine
Sediments. *PLoS One* **9**, e101443 (2014).
- 100. Louca, S., Wegener Parfrey, L. & Doebeli, M. Decoupling function and taxonomy in the global ocean
microbiome. *Science (80-.)*. **353**, 1272–1277 (2016).
- 101. Gomes, N. C. M., Cleary, D. F. R., Calado, R. & Costa, R. Mangrove bacterial richness. *Commun. Integr.*
*Biol.* **4**, 419–423 (2011).
- 102. Cabral, L. *et al.* Anthropogenic impact on mangrove sediments triggers differential responses in the
heavy metals and antibiotic resistomes of microbial communities. *Environ. Pollut.* **216**, 460–469 (2016).
- 103. Takahashi, S., Tomita, J., Nishioka, K., Hisada, T. & Nishijima, M. Development of a prokaryotic
universal primer for simultaneous analysis of Bacteria and Archaea using next-generation sequencing.
*PLoS One* **9**, e105592 (2014).
- 104. Klindworth, A. *et al.* Evaluation of general 16S ribosomal RNA gene PCR primers for classical and next-
generation sequencing-based diversity studies. *Nucleic Acids Res.* **41**, e1–e1 (2013).
- 105. Martin, M. Cutadapt removes adapter sequences from high-throughput sequencing reads.
*EMBnet.journal* **17**, 10–12 (2013).
- 106. Callahan, B. J. *et al.* DADA2: High-resolution sample inference from Illumina amplicon data. *Nat.*
*Methods* **13**, 581–583 (2016).
- 107. Callahan, B. J., McMurdie, P. J. & Holmes, S. P. Exact sequence variants should replace operational
taxonomic units in marker-gene data analysis. *ISME J.* **11**, 2639–2643 (2017).
- 108. Quast, C. *et al.* The SILVA ribosomal RNA gene database project: improved data processing and web-
based tools. *Nucleic Acids Res.* **41**, D590–D596 (2012).
- 109. McMurdie, P. J. & Holmes, S. phyloseq: An R Package for Reproducible Interactive Analysis and
Graphics of Microbiome Census Data. *PLoS One* **8**, e61217 (2013).
- 110. Wright, E. S. Using DECIPHER v2.0 to analyze big biological sequence data in R. *R J.* **8**, 352–359 (2016).
- 111. Schliep, K. P. phangorn: phylogenetic analysis in R. *Bioinformatics* **27**, 592–593 (2011).
- 112. Kembel, S. W. *et al.* Picante: R tools for integrating phylogenies and ecology. *Bioinformatics* **26**, 1463–
1464 (2010).
- 113. Faust, K. & Raes, J. Microbial interactions : from networks to models. *Nat. Publ. Gr.* **10**, 538–550 (2012).
- 114. Weiss, S. *et al.* Correlation detection strategies in microbial data sets vary widely in sensitivity and
precision. *ISME J.* **10**, 1669–1681 (2016).
- 115. Faust, K. & Raes, J. CoNet app : inference of biological association networks using Cytoscape [version 2;
referees: 2 approved]. *F1000Research* 1–16 (2018) doi:10.12688/f1000research.9050.1.
- 116. Bastian, M. & Heymann, S. Gephi: An Open Source Software for Exploring and Manipulating Networks.
(2009).
- 117. Barberán, A., Bates, S. T., Casamayor, E. O. & Fierer, N. Using network analysis to explore co-
occurrence patterns in soil microbial communities. *ISME J.* **6**, 343–351 (2012).
- 118. Assenov, Y., Ramírez, F., Schelhorn, S. E. S. E., Lengauer, T. & Albrecht, M. Computing topological
parameters of biological networks. *Bioinformatics* **24**, 282–284 (2008).
- 119. Van Rossum, G. & Drake, F. L. The Python Language Reference — Python 3.7.4 documentation.
<https://docs.python.org/release/3.7.4/reference/index.html> (2019).

- 120. Core Team, T. R: A language and environment for statistical computing. R Foundation for Statistical
Computing, Vienna, Austria. (2016).
- 121. Oksanen, J. *et al.* vegan: Community Ecology Package. R package version 2.4-3. (2016).
- 122. Wickham, H. *ggplot2: Elegant Graphics for Data Analysis*. (Springer, 2009).
- 123. Clarke, K. & Gorley, R. PRIMER v6: User manual/tutorial. *Prim. Plymouth* 192 (2006).

**Figure legends and tables**

**Figure 1 Within sample diversity of bacterial communities from arid mangrove soils**

Alpha diversity measures of the bacterial communities detected in mangrove soils from Australia
(green outline) and Saudi Arabia (orange outline) across zones (fringe, shrub) and depths (surface in
light grey, subsurface in dark grey). **A** Rank-abundance relationship of ASVs by location. **B** Observed
ASVs. **C** Shannon diversity index. **D** Phylogenetic diversity index (Faith's PD). The boxplots indicate
the median with the interquartile range (IQR) between the 25th and the 75th percentile and the
whiskers extend $1.5 \cdot \text{IQR}$. They were plotted with the notch ($\pm 1.58 \cdot \text{IQR} / \sqrt{n}$) to display likely
statistical significance.

**Figure 2 Comparison of different bacterial communities between the experimental factors** 919 **geographic region, exposure, zone, and depth**

**A** Relative abundance of phyla across the whole data set. The top 12 most prevalent phyla were
displayed and the remaining grouped as 'Other'. **B** Schematic representation of the results from the
variance partitioning. The large bubble represents the proportion of variation explained by the four-
way interaction of all experimental factors. The small bubbles show the proportion of variation
explained by each single factor. The arrow above indicates the environmental factors influencing the
separation. **C** Non-metric multidimensional scaling (NMDS) plots of the community matrix across
experimental factors at ASV level, and **D** at genus level across geographic region, exposure, zone and
depth.

**Figure 3 Bacterial co-occurrence networks of each site in Australia and Saudi Arabia across** 930 **exposure, zone and depth**

The colours denote different modules within each network but do not show connections between
networks per se. The size of the nodes is relative to their node degree.

**Figure 4 Selected network metrics by phylum**

**A** Betweenness centrality and **B** Topological coefficient. The 12 most prevalent phyla (within the
networks) were selected and the remaining grouped as 'Other'. Mean scores are displayed with
standard errors calculated by the `stat_summary` function in `ggplot`.

**Figure 5 Comparison of high degree scores in co-occurrence networks**

Density plots of the relative proportion of nodes against the normalized node degree coloured by
 phylum across geographic region, exposure, zone and depth. The proportion of nodes is calculated
 by the Kernel-Density function.

**Figure 6 Theoretical functional assignments**

Heat map showing the distributions of bacterial functions that were assigned by FAPROTAX across
 geographic region, exposure, zone and depth. Values were log-transformed with lighter values
 indicating higher abundances.

**Figure 7 Map and sampling design of the nested factors**

World map showing the global regions of the sampling sites on the Western Australian coast and in
 Saudi Arabia. The more detailed maps show the exposed and the sheltered sites in both geographic
 regions. An aerial photograph visualizes the separation between the zones of each forest, with
 actual sampling locations at the sheltered site in Saudi Arabia separated into three plots each at the
 tall fringe and the shrub. The schematic representation of the sampling design below, shows the
 arrangement of the experimental factors and the numbers of replicates taken. The values in brackets
 represent the number of replicates for each sample. **The numbers in the last row denote the**
 **sampling depth in centimetres.** Asterisks (*) represents from which site a sample that has been
 removed for insufficient sequencing depth or cross-contamination, making the total number of
 samples n = 115.

**Table 1** Summary statistics characterizing the bacterial co-occurrence networks of all sites in
 Australia and in Saudi Arabia.

Geographic region	Exposure	Zone	Depth	Nodes	Edges	Edges per Node	Clustering Coefficient	Centralisation	Diameter	Average Neighbors	Density	Number of Modules
Australia	Exposed	Fringe	Surface	609	14793	24.29	0.58	0.52	7	48.58	0.08	6
			Subsurface	609	14793	24.29	0.58	0.52	7	48.58	0.08	6
		Shrub	Surface	451	6504	14.42	0.7	0.11	9	28.84	0.06	14
			Subsurface	597	8333	13.96	0.7	0.6	7	27.92	0.05	5
	Sheltered	Fringe	Surface	495	5808	11.73	0.62	0.67	6	23.47	0.05	5
			Subsurface	733	6323	8.63	0.44	0.4	7	17.25	0.02	6
		Shrub	Surface	675	2024	3	0.45	0.02	21	6	0.01	17
			Subsurface	418	5530	13.23	0.55	0.36	6	26.46	0.06	3
Saudi Arabia	Exposed	Fringe	Surface	423	9642	22.79	0.52	0.45	7	45.59	0.11	3
			Subsurface	736	5387	7.32	0.33	0.16	10	14.64	0.02	10
		Shrub	Surface	737	6088	8.26	0.37	0.26	7	16.52	0.02	10
			Subsurface	769	5514	7.17	0.37	0.29	8	14.34	0.02	8

	Sheltered	Fringe	Surface	726	5569	7.67	0.32	0.43	10	15.34	0.02	7
			Subsurface	661	4178	6.32	0.42	0.24	11	12.64	0.02	9
		Shrub	Surface	609	4812	7.9	0.38	0.22	8	15.8	0.03	7
			Subsurface	736	5725	7.78	0.3	0.31	11	15.56	0.02	9

963

Geographic Region

 Australia
 Saudi Arabia

Zone

 Fringe
 Shrub

Depth

 Subsurface
 Surface

Geographic Region

- Australia
- Saudi Arabia

Exposure

- Exposed
- Sheltered

Zone and Depth

- Fringe Subsurface
- Fringe Surface
- △ Shrub Subsurface
- ▲ Shrub Surface

Australia

Fringe

Shrub

Saudi Arabia

Fringe

Shrub

Exposed

Surface

Subsurface

Sheltered

Surface

Subsurface

Phylum

Acidobacteria
Actinobacteria

Bacteroidetes
Calditrichaeota
Chloroflexi

Cyanobacteria
Gemmatimodetes
Nitrospirae

Other
Planctomycetes
Proteobacteria

Spirochaetes
Zixibacteria

$\log(\text{Abundance} + 1)$

0 2 4 6

Assigned Function

Response to the Editor

Can the authors please explain how they define structure versus composition in this study. Depending on the study system or ecological question being tested, these terms may be synonymous. It seems as though the authors define structure differently and refer to the "structure" as network topology characteristics and composition as other aspects of community alpha and beta diversity.

We want to thank the editor for this additional recommendation and agree that the subtle differences in definitions are better clearly explained to avoid confusion. To clarify, we have added the following sentence to the introduction in L112-114:

“Throughout this paper, we refer to community structure as network topology characteristics that imply interactions within the community, and we define community composition as alpha and beta diversity.”

Response to the reviewers

Reviewer #1 (Comments for the Author):

It has been a pleasure for me to review this manuscript. The authors present a very well designed study to investigate the effects of local environmental and biogeographic factors on mangrove soil microbial communities. The methodology is well described, their approach scientifically sound, and the conclusions well supported by the data.

I only have a few minor comments and suggestions:

Throughout the manuscript, the authors are referring to 'abundance' of microbes and their functions. However, as their conclusions are only based on compositional sequence data, I strongly recommend to rephrase and use the term 'proportion' instead. Or clearly define at the beginning that any statements about abundance are only referring to sequence proportions.

We agree with the reviewer that the data we present in this manuscript refers to the proportion of bacterial sequences in each sample. The word abundance is used in varying ways throughout the manuscript. When it was used in regards to the proportion of microbes, it is always preceded by the term 'relative', which implies it being a proportional measure (Examples in L191-197). In my understanding, 'relative abundance' is a well-established term to describe proportions of microbial sequences within a sample. An explanation has been added in L187. In the discussion of the network analysis the term is used twice in a theoretical concept (L443-446) and should remain as it is.

For the interpretation of the network analysis, I would like to suggest more caution in equating network topology with ecological importance. Co-occurrence networks may rather be used to develop hypotheses about ecological interactions, than to actually prove those interactions.

L168: Is this 'formula' referring to the interaction term? If yes, I would not describe an interaction as 'consistent effect', but rather use a similar description as provided in L153.

The description of the interaction term was changed to match the previous descriptions L174-176.

L194: Most likely a typographical error, but what is the a) after the inner parentheses referring to?

This was indeed a typo and has been removed (L211).

L602: How does the 30% of ASVs used for the functional prediction translate to total sequence proportions? Do you think that the functional assignment may be biased due to differing proportions of uncharacterized taxa (not part of the FAPROTAX table) in different samples?

I agree that this is a very relevant question for the use of functional annotation tools in general. FAPROTAX bases its assignments only on the culturable fraction of the sequences, which provides a high level of confidence, that the assigned functions can actually be carried out by the present community. This is different to other tools like paprica or tax4fun, which use known metagenomic profiles for metabolic inference. Therefore, it can be expected that the proportion of assigned sequences to functions would be larger with another tool. However, this will be at the cost of confidence in the predictions made.

Functional assignments come with strong biases, especially when a large proportion of taxa are uncharacterized.

L603: What was your reason for excluding collinear functions from the analysis?

We agree that this is not described very clearly. We were interested in the dominant functional groups that represented different metabolic pathways within the community. FAPROTAX covers several similar pathways (i. e. "respiration of sulphur compounds" and "sulphate respiration", or "chemoheterotrophy" and "aerobic chemoheterotrophy"). To avoid overloading of the already well populated analysis of inferred functions, redundant functions which aligned collinearly, were removed. The functions that were kept were the ones that were more general (chemoheterotrophy over aerobic chemoheterotrophy).

We have added an explanation of this in L644-646.

Submitted sequence data: As mentioned in the separate field above, I suggest to modify the metadata for the following parameter:

- library_source should be METAGENOMIC

Done

- environmental_sample should be TRUE

The data is specified as environmental sample on the summary page of the BioProject under 'Relevance'. This is slightly different as compared to the meta data at ArrayExpress, however the information requested is openly available.

<https://dataview.ncbi.nlm.nih.gov/object/PRJNA720541>

- target_gene is missing and should be 16S rRNA

Again, NCBI's SRA database has a different layout for their meta-data and target_gene cannot be specified. However, the column 'Library strategy' in the experiment information states 'AMPLICON' and a detailed description of the specific primers used and the target gene and region amplified is given in the 'Experiment description'.

<https://dataview.ncbi.nlm.nih.gov/object/SRR14181447>

Reviewer #2 (Comments for the Author):

The manuscript by Thomson et al. presents the bacterial community associated with the soil under *Avicennia marina* in distinct geographical areas and in subareas of local importance (sheltered/exposed; fringe/shrub). The question is relevant and clearly exposed. The use of molecular and bioinformatics tools allowed to have a satisfactory description of the studied communities. However, the absence of physical and chemical data or other environmental data, such as temperature, rainfall, and evapotranspiration (mostly important for arid mangroves) is an important flaw. I consider this paper will be helpful however there are some concerns to address. The data analysis is robust but the authors should try to organize results and discussion to improve clarity. I have also recommended including some discussion topics for better contextualization.

L8: All superscript from the affiliations are too small.

The size of all superscripts has been adjusted to match the text size of the document.

In line with this comment, all subscripts were also enlarged (degrees of freedom in statistical reporting)

L32: I suggest changing to " High-throughput amplicon sequencing was applied..."

The text was changed as suggested L33.

L33-34: Consider changing to regional.

The text was changed as suggested L35.

L34: no comma

Done L36

L53: no comma.

Done L55

L55: no comma.

Done L56

L56: add comma.

Done L58

L87-90: In fact, there are many published studies on the effect of local conditions on the mangrove soil microbiome.

Some examples: presence or absence of vegetation (Jiang et al., 2013; Gomes et al., 2014; Prakash et al., 2015; Chen et al., 2016; Liu et al., 2017), depth (Taketani et al., 2010; Andrade et al., 2012; Otero et al., 2014; Basak et al., 2016; Luis et al., 2019), human activities (Dias et al., 2011; Peixoto et al., 2011; Fernandes et al., 2014; Nogueira et al., 2015; Basak et al., 2016; Erazo and Bowman, 2021), tidal cycles (Zhang et al., 2018), and physicochemical properties (organic carbon and nitrogen (Chen et al., 2016), ammonia nitrogen (Zhu et al., 2018b), and silt-clay content (Colares and Melo, 2013)).

We thank the reviewer for the thorough suggestion of references relating of the distribution of microbes in mangrove soils and agree that the indicated sentence doesn't give enough credit to these published efforts. The sentence was modified to include some references and convey with more clarity the argument the authors were trying to make L115-117.

The recent paper by Tavares et al. 2021 was also included.

We acknowledge the efforts made to understand factors driving local community composition but want to highlight the need for a more comprehensive approach over multiple spatial scales and to understand the mechanisms underlying microbial community assembly.

L101-103: Please see the papers from Li et al. 2021 and Zhang et al. (2021).

Li et al. (2021). Active bacterial and archaeal communities in coastal sediments: Biogeography pattern, assembly process and co-occurrence relationship. doi:10.1016/j.scitotenv.2020.142252.

Zhang et al. (2021). Biogeography, Assembly Patterns, Driving Factors, and Interactions of Archaeal Community in Mangrove Sediments. doi:10.1128/mSystems.01381-20.

I am very grateful for these two papers as I had not seen them before the submission of this manuscript. I agree that they both add some valuable insights to the discussion of microbial community assembly in coastal areas. The novelty of the study presented here lies rather in the experimental design which uses a hierarchical design with the sites 'stacked' within each other.

L104-107

The two references were also added to the discussion L332 and L334.

L115-117: Consider removing this fragment.

The sentence was removed L122-124.

L143-144: Please, add information on the F value and P for this test, as presented in Supplem Table s4. Also, indicate this difference in the corresponding box-plot image.

To improve clarity in the presentation of the results, the reference to the statistical outcomes was placed at the beginning of each paragraph (L152-154, L167-168, L174-176). As we have complex significant interaction terms, only the highest level of complexity is presented (highest number of interacting factors).

Since we have a three-way interaction it is very confusing to add letters or symbols to the boxplot to indicate significance. The notch which we use in the boxplots presents an elegant solution to this problem. The notch is calculated as: $\text{median} \pm 1.58 \cdot (\text{interquartile range} / \sqrt{n})$ <- this is noted in the figure caption L978. At 95% confidence, if the notches don't overlap, it signifies a likely significant difference between the medians.

L149: Please, add information on the F value and P for this test, as presented in Supplem Table s4 ou s6. Also, indicate this difference in the corresponding box-plot image.

Please see the reply to the comment above.

L149-152: This result can be observed in Figure 1C and 1D?

This result is still in reference to Figure 1B. To clarify, I have added an insert in L154 to connect species richness with the number of observed ASVs. Species richness is the ecological parameter which is used and the number of observed species is the metric in which it is recorded, hence the y-axis description in figure 1B. I have also added an explanation of this in the figure legend in L974-975. I have also added a reference to figure 1B and the anova table in the supplementary materials S4 in L154.

The results that can be seen in Figures 1C and 1D are discussed in the following paragraphs, L167-168, and L174-184, respectively.

Consider separating in different paragraphs the results of each of studied factors to improve clarity.

We agree that the section would benefit from a more structured layout. To improve the clarity of separation between factors, I have added a paragraph in L150.

Also, this statement is very strange to me as I could not observe such differences in Figure 1. Maybe it is because the color code for surface and subsurface are switched when comparing Figure 1 and Supp. Figure S5. Plase, revise this carefully because it is quite confusing.

Thank you very much for pointing out that the legend does not match the figure. The figure remains as it stands but the labels for depth in the legend have been switched as a result of the graphical rearrangement. It is correct in the supplementary file (Figures S5, S8, and S10) with the light grey boxes visualizing the surface and the dark grey boxes the subsurface diversities.

The changes to Figure 1 have been made.

Also, check carefully the analysis of variance results because a small detail of the F value present in Line 159 do not match.

I commend the reviewer's wonderful attention to detail and have rectified the discrepancy to match the anova table in the supplementary materials (Supplementary table S6). L168

L172: Consider including a subsection to present results on Composition

A sub-header was inserted in L186, leading into a subsection on the composition of the bacterial communities

L172: Correct to "was".

Done L188

L180-182: This is true for the shrub samples, but seems a little forced for the fringe samples. Please, correct the fragment.

The sentence in question has been modified to describe the differences between surface and subsurface samples more accurately (L196-197).

L186: Change "dominant" to "abundant".

Done L203

L187-188: Please, add the exposed sites in Saudi Arabia also present more elevated abundances than the sheltered sites.

Please, include discussion on this issue at the Discussion section.

Reference to the exposed site in Saudi Arabia was added in L 204-205.

This was added to the discussion in L411-415.

L208-209: Did the depth also fail to separate the data in the ordination space? Please, include the results for this factor regarding Figure 2D.

The sentence has been amended to include information on the effect of depth on the sample distribution in the NMDS plot L226-228.

L233-252: The description of the results is quite confusing. I suggest re-writing those fragments to improve clarity.

A suggestion is to organize results for betweenness centrality and topological coefficient by site and zone. In this way, the reader can visualize better who are the phyla that better relates to each factor. Also, I think that the polar graphs chosen to show these results are not facilitating the visualization of the phyla present and in some grids it is almost impossible to discern between the groups.

We agree, that it is hard to follow the description of the results, mainly induced by the multiple levels of comparison. We therefore followed the reviewer's advice and rearranged the paragraphs by experimental factor and not by phylum L252-257 and L269-272.

The polar graphs have been changed to bar graphs (Figure 4).

L281-285: The authors should re-think or explain better this assumption because there are geographical differences in the bacterial communities between Saudi Arabia and Australia, as the authors themselves discuss in lines 341-356. Also, geographical region appeared as an important factor in many tests presented in the manuscript.

We agree that the subtleties of the analysis need to be explained carefully to convey the point and the main finding of the study. We have therefore modified the first paragraph of the discussion (L309-313) to emphasise that both factors, geographic region and forest zone have an influence on the bacterial community composition of the mangrove forest, but highlight, that the local factor is stronger in doing so. We then flow into the discussion in which we discuss the local factors and then the geographic factors in turn. We mention in these sections the different tests, that highlighted the importance of both factors.

L287: No comma.

Done L315

L309-310: It is possible to see this "pattern" of higher diversity in fringe areas in Australia and higher diversity in shrub areas in Saudi Arabia. What might be the reason of such difference?

This is a good point to consider and we have included a sentence regarding this question in relation to the proposed stress gradient hypothesis in L338-340.

L361: Remove "between".

Done L391

L372: Are those phyla really dominant? Consider changing the word "dominant" for another one that refers better to the role of those phyla in the constructed networks.

The word dominant was exchanged with one that better highlights their roles in the network structure in L402.

L422-424: The authors can discuss better this inference. Please, see the Stress Gradient Hypothesis (by Bertness and Callaway, 1994; see Hernandez et al., 2021 for a better explanation on the effects on microbial communities).

See also the Intermediate Disturbance Theory (Padisak, 1993).

doi:10.1016/0169-5347(94)90088-4

doi:10.1038/s41396-020-00882-x

doi:10.1007/978-94-017-1919-3

We thank the reviewer for the suggestion of this discussion point and added the two theories and the references to the discussion. A note to how this may relate to our study was also included here L457-461.

L460: No comma.

Done L497

L462: Change to "carbon sequestration and nutrient processing".

The change has been made as requested (L499).

Site descriptions and experimental design:

L485-497: Improve sampling description because it is very confusing. As you showed in the drawing in Figure 7, describe the division of the sampling areas accordingly.

The paragraph in question was revised. To improve clarity and structure, the sentence on the sampling dates was moved to the preceding paragraph 516-517.

The sampling areas were described in order from large scale to small scale, as in the graphic in figure 7, L518-526.

The description of the sampling procedure was changed L527-531.

L487-490: Why did the authors collected six cores in Australia and nine in Saudi Arabia? Please, explain and detail this in the section. Also, I do not think the description given here matches with the illustrated description presented in Figure 7. Please, verify and/or explain.

A brief explanation for why sample number were different in the two countries was added to the text L528-529. A line of text was added to match the description of the number of samples collected and the description in Figure 7 (L539-540).

A bit more detail here: Discrepancies in sample numbers from the two locations are due to resource limitation in Australia. The timing and coordination of this trip was limited due to fieldwork logistics of the organizing group.

L487: Add "respectively".

I believe that this does not longer apply as the sentence has been changed for more clarity.

L554: Please, improve the readability of the rarefaction curve presented in the Supplementary File. Even if it is a supplementary file, it is supposed to be readable. Removing gridlines and increasing font to size might be enough.

The horizontal sample lines were removed from the graph. All efforts to make the labels larger resulted in substantial overlap of the labels not improving the issue. Eventually the labels were removed all together (Supplementary figure 2).

L606: The authors did not present physical and chemical data or other environmental data, such as temperature, rainfall, and evapotranspiration (mostly important for arid mangroves). Did the authors not have access to this type of data?

We agree with the reviewer that the inclusion of environmental, physical, and chemical data would have been preferable and greatly benefitted the study. Unfortunately, due to limited time and logistics of the field trip in Australia, we were not able to collect associated environmental data from the Australian sites. This is unfortunate and in hindsight should have been given more priority.

Larger scale climate data (annual temperature, rainfall, and tidal amplitude) was reported in the methods section (L511-515).

L608: No comma.

Done L651

L649: Correct to KAUST.

Done L693

L956-957: Please, add some text in parenthesis explaining that <2 are samples from surface and -10 are samples from subsurface. Another option is to change this information in Figure 7 to surface and subsurface instead of <2 and -10.

Done L101-1020.

Table 1: The font size is too small. Increase it for a better readability.

Also, standardize how many numbers will be presented after the comma.

Yes, we agree, the table has been shifted to a landscape layout, font size increased and numbers all standardised within each column. (L1024-1025)

Figure 1: Check for the color pattern for surface and subsurface. In the supplementary info there is a color code which is different from this one. This is causing some trouble in understanding the results related to this figure.

We thank the reviewer for picking up on this minute but essential detail. The legend has been changed accordingly (Figure 1).

Figure 7: Identify in the map Saudi Arabia and Australia. Also, reduce the size of the dots in the map indicating the sampling points to better visualize how separated they were. Why did you include a satellite zoom from Saudi Arabia but not for Australia? Please, if possible include satellite zoom for Australia.

We thank the reviewer for the improving suggestions to the Figure 7. The geographic regions of Saudi Arabia and Australia have been identified on the map.

We agree with the reviewer that the dots are not distinguishable between the two locations in the panel showing the exposed and sheltered locations of the two countries. The maps were made with the original coordinates. The scale of the map displayed is too small to visualise the two dots as individual dots without manually removing and artificially superimposing them. The joint dots have been replaced by single dots indicating the forest location. For a higher separation of the sites, we included an example of one detailed site description below (sheltered site in Saudi Arabia).

We included the aerial photograph of the sheltered zone in Saudi Arabia as an example for all four zones, so that we don't need to show all four sites. This is explained in the caption by "An aerial photograph visualizes the separation between the zones of each forest, with actual sampling locations at the sheltered site in Saudi Arabia separated into three plots each at the tall fringe and the shrub" but we now added "as an example for sampling separation within each of the four zones.", to further clarify this L1016.

The description of the depths has been changed from <2 and -10 to 'Surf' and 'Sub', respectively Figure 7 and L1025-1026.

November 1, 2021

Mr. Timothy Thomson
University of Waikato
Tauranga
New Zealand

Re: Spectrum00903-21R1 (Contrasting effects of local environmental and biogeographic factors on the composition and structure of bacterial communities in arid monospecific mangrove soils)

Dear Mr. Timothy Thomson:

Thank you for submitting your manuscript to Microbiology Spectrum. I have reviewed your revised manuscript and would like an additional set of revisions based on recommendations given below. If the authors thoroughly address these comments, I am fully supportive of accepting this manuscript for publication. Please see my comments below.

When submitting the revised version of your paper, please provide (1) point-by-point responses to the issues raised by the reviewers as file type "Response to Reviewers," not in your cover letter, and (2) a PDF file that indicates the changes from the original submission (by highlighting or underlining the changes) as file type "Marked Up Manuscript - For Review Only". Please use this link to submit your revised manuscript - we strongly recommend that you submit your paper within the next 60 days or reach out to me. Detailed information on submitting your revised paper are below.

Link Not Available

Sincerely,

Allison Veach

Journals Department
Editor Comments:

Introduction

Line 70: repetitive, state "... additional ecosystem services, microbial communities have not received the same attention as those in other environments...."

Line 75: anything that is decomposing will be a hotspot as decomposing is a microbial activity. Can the authors say "Notably, leaf and root litter are hotspots of microbial activity due to decomposition."

Line 95-101: These are not necessarily contrasting theories. The first describes only environmental selection as being important and the other recognizes that both environmental selection (niche-based processes) and stochasticity (neutral dynamics) impact microbes. Can the authors rewrite this paragraph so that you are describing the latter, community assembly theory, in more detail and not bring up the Baas-Becking hypothesis? This would link this work more so in a general community ecology context.

Methods

L517: Should it say 249mm and 54mm? Is there a digit missing?

Results

L149: Is this effect seen across both regions and for exposed versus sheltered? Be explicit in what the rank abundance curves indicate.

Throughout the results, can the authors please state the average proportional differences in richness, H', and Faith's PD between significant levels of a fixed effect? Its repeatedly said a site is higher or lower than another, but that doesn't indicate the magnitude of that effect. Please do this throughout the results to help readers understand which effects may be more influential.

L154 and throughout results: please add Tukey HSD corrected p-values when discussing statistical differences.

L160: It states that depth did not influence ASV richness, but here it states it does. Please correct.

L178: What is meant by inconsistent? Do the authors refer to highly variable?

L188: Please break down Proteobacteria in sub-phyla or classes (i.e., Alpha-, Betaproteobacteria, etc) and describe which of those groups changed within your study design. Change Figure 2A accordingly.

Line 236: Why do the authors use perMANOVA but do not show R2 values which also indicate the amount of variation explained by a factor? The 'varpart' function is quite similar but uses RDA as its ordination method instead of NMDS. I suggest either retaining varpart-variance partitioning and show an RDA instead of an NMDS, or just do an NMDS and use perMANOVA to determine amount of variation per factor.

L257: What differences were demonstrated with betweenness scores for Cyanobacteria and Nitrospira? There is a lot of useful information for this metric and in Figure 4, yet the authors do not provide very much information regarding important phyla for centrality. Can the authors please provide a more systematic description of what phyla are important for each factor in this study?

Discussion

L312: should write "based on alpha and beta diversity."

L313-316: This sentence is very confusing. Can you please restate this?

L319: composition and co-occurrence are intrinsically linked and there cannot be a disconnect from each other. They both varied by geographic region. Do co-occurrence networks take into account relative abundance or is it a presence/absence-based analysis? That may be why you see different factors mattering more so for one versus another.

L339: What is energetic referring to? Please explain.

L353: grouped, not group

L358: It is highly speculative to state dispersal limitation as your analyses don't actually test this. Please scale back this statement.

L360-367: Similarly, your analysis does not test evolutionary factors. Please scale back.

L418: Capitalize phylum names.

L419: network topology, not network community.

L502: I suggest stating that this work supports the importance of environmental selection, not universal rules.

Tables/Figures

Figure1A - can the authors make the points easier to read? Less transparency or a different color? It's difficult to see the few dominant taxa.

Please place Figure 3 in supplemental material. There are too many figures and the network topologies are not particularly easy to digest and useful in interpretation. In addition, what do "modules" represent again? Its not clear what color and what each point represents.

Staff Comments:

Preparing Revision Guidelines

Please return the manuscript within 60 days; if you cannot complete the modification within this time period, please contact me. If you do not wish to modify the manuscript and prefer to submit it to another journal, please notify me of your decision immediately so that the manuscript may be formally withdrawn from consideration by Microbiology Spectrum.

Editor Comments:

Introduction

Line 70: repetitive, state "... additional ecosystem services, microbial communities have not received the same attention as those in other environments...."

L68-69: The sentence in question was amended as suggested.

Line 75: anything that is decomposing will be a hotspot as decomposing is a microbial activity. Can the authors say "Notably, leaf and root litter are hotspots of microbial activity due to decomposition."

L73-74: Amended accordingly.

Line 95-101: These are not necessarily contrasting theories. The first describes only environmental selection as being important and the other recognizes that both environmental selection (niche-based processes) and stochasticity (neutral dynamics) impact microbes. Can the authors rewrite this paragraph so that you are describing the latter, community assembly theory, in more detail and not bring up the Baas-Becking hypothesis? This would link this work more so in a general community ecology context.

L92-104: The paragraph has been changed to describe the species assembly concept.

Methods

L517: Should it say 249mm and 54mm? Is there a digit missing?

It is correct as it is written in the manuscript. Saudi Arabia experiences a hyper arid climate and the region around Jeddah receives a mean of 54 mm of rain annually (in line citations are provided in L512).

Results

L149: Is this effect seen across both regions and for exposed versus sheltered? Be explicit in what the rank abundance curves indicate.

L149-151: Two sentences were included to describe differences of the rank abundance curves between the samples.

Throughout the results, can the authors please state the average proportional differences in richness, H' , and Faith's PD between significant levels of a fixed effect? Its repeatedly said a site is higher or lower than another, but that doesn't indicate the magnitude of that effect. Please do this throughout the results to help readers understand which effects may be more influential.

L157, 159, 161, 168, 174: proportional differences of the reported significances were added.

L154 and throughout results: please add Tukey HSD corrected p-values when discussing statistical differences.

L157, 159, 161, 168, 174: Done.

L160: It states that depth did not influence ASV richness, but here it states it does. Please correct.

L162-164: We agree. The sentence has been removed from the manuscript.

L178: What is meant by inconsistent? Do the authors refer to highly variable?

L172-174: This observation was meant to point out the difference between the zones of the sheltered site in Australia compared with all other sites. At the exposed site in Australia as well as both sites in Saudi Arabia, the phylogenetic diversity was higher in the shrub zone than in the fringe.

We agree, that this is not communicated properly and have changed the sentences in question to be more descriptive.

L188: Please break down Proteobacteria in sub-phyla or classes (i.e., Alpha-, Betaproteobacteria, etc) and describe which of those groups changed within your study design. Change Figure 2A accordingly.

Fig 2A: Changes were made to the figure as requested.

L192-194: A description of differences between *Delta*- and *Gammaproteobacteria* was included in the text.

L986-990: This change was also mentioned in the figure caption.

Line 236: Why do the authors use perMANOVA but do not show R2 values which also indicate the amount of variation explained by a factor? The 'varpart' function is quite similar but uses RDA as its ordination method instead of NMDS. I suggest either retaining varpart-variance partitioning and show an RDA instead of an NMDS, or just do an NMDS and use perMANOVA to determine amount of variation per factor.

L235-238: We agree that using both contrasting ordinations is redundant. The varpart analysis was removed (L203-212) and the R2 values were added to the text.

L257: What differences were demonstrated with betweenness scores for Cyanobacteria and Nitrospira? There is a lot of useful information for this metric and in Figure 4, yet the authors do not provide very much information regarding important phyla for centrality. Can the authors please provide a more systematic description of what phyla are important for each factor in this study?

L251-170: We agree that the description of the network metrics was not very detailed. We have added a few sentences to highlight the various Phyla that play a role in differentiating the factors. However, we want to note that there don't seem to be any dominating trends across all factors, which would make this much clearer.

The structure of these paragraphs was changed in the previous revision to systematically describe differences between our experimental factors, as this was requested by Reviewer 2.

Discussion

L312: should write "based on alpha and beta diversity."

L305: The addition was made and the sentence was split into two to avoid creating an overly long sentence.

L313-316: This sentence is very confusing. Can you please restate this?

L307-309: We agree that the sentence in question needed rewriting. It was simplified to make its intention clearer.

L319: composition and co-occurrence are intrinsically linked and there cannot be a disconnect from each other. They both varied by geographic region. Do co-occurrence networks take into account relative abundance or is it a presence/absence-based analysis? That may be why you see different factors mattering more so for one versus another.

The network correlation matrix is calculated using various different metrics (two distance based and two correlation based) on the same abundance table as the community composition, so they are linked. Relative abundance of taxa is considered by both distance-based metrics. The method used is fairly conservative though which makes us confident in the differences we see between the networks and the compositional data. This means that while there is an intrinsic relationship between composition and co-occurrence, their dissimilarities as presented here are to be considered.

L310-311: We agree that the wording was too strong and have amended the sentence in question.

L339: What is energetic referring to? Please explain.

L331: The higher energy environment at the fringe through wind wave and tidal movement, that leads to increased homogenization of the upper layers of the sediment.

We agree that this is not clear and have changed it to "... away from the homogenizing forces of the fringe..."

L353: grouped, not group

L345: Agreed and amended.

L358: It is highly speculative to state dispersal limitation as your analyses don't actually test this. Please scale back this statement.

L348-352: We agree that dispersal limitation was not tested in this study. To match that, the sentences discussing this effect have been toned down and modified to make clear that this is a theoretical embedding of our findings within the existing literature.

L360-367: Similarly, your analysis does not test evolutionary factors. Please scale back.

L352 & 359-360: As above, the sentences have been scaled back. It was made clearer that these are hypothetic statements and not conclusions from our findings. The sentence suggesting this has been deleted from the manuscript (L335-336).

L418: Capitalize phylum names.

L411: Capitalized and italicized.

L419: network topology, not network community.

L412: Thank you, done.

L502: I suggest stating that this work supports the importance of environmental selection, not universal rules.

L492-496: The sentence was changed as suggested. The following sentence was modified to avoid repetition of words and structure.

Tables/Figures

Figure 1A - can the authors make the points easier to read? Less transparency or a different color? It's difficult to see the few dominant taxa.

Figure 1A: The figures have been changed. The symbols have been slightly enlarged and the grey has been made darker for a stronger contrast to the background. Lines were added to better describe the curves of each sample. We thank the editor for this suggestion, as the figure is much easier to read now. The colour (grey-scales) was retained, as it matched the depth colours of the neighbouring box plots.

Please place Figure 3 in supplemental material. There are too many figures and the network topologies are not particularly easy to digest and useful in interpretation. In addition, what do "modules" represent again? Its not clear what colour and what each point represents.

The Network figure was moved to the supplementary materials as an individual file (Supplementary file 2).

L247: reference to the figure was changed.

L618-621: A sentence was added to the methods explaining modules within a network.

December 11, 2021

Mr. Timothy Thomson
University of Waikato
Tauranga
New Zealand

Re: Spectrum00903-21R2 (Contrasting effects of local environmental and biogeographic factors on the composition and structure of bacterial communities in arid monospecific mangrove soils)

Dear Mr. Timothy Thomson:

I'm glad to inform you your manuscript has been accepted. I am forwarding it to the ASM Journals Department for publication. You will be notified when your proofs are ready to be viewed.

Sincerely,

Allison Veach
Editor, Microbiology Spectrum

Journals Department
Supplemental Information workflow: Accept